# MOF-mediated histone H4 Lysine 16 acetylation governs mitochondrial and ciliary functions by controlling gene promoters

Dongmei Wang [1,2], Haimin Li[1], Navdeep S. Chandel [2,3], Yali Dou [4] & Rui Yi [1,2,5] ✉

Histone H4 lysine 16 acetylation (H4K16ac), governed by the histone acetyltransferase MOF, orchestrates gene expression regulation and chromatin interaction. However, the roles of MOF and H4K16ac in controlling cellular function and regulating mammalian tissue development remain unclear. Here we show that conditional deletion of *Mof* in the skin, but not *Kansl1*, causes severe defects in the self-renewal of basal epithelial progenitors, epidermal differentiation, and hair follicle growth, resulting in barrier defects and perinatal lethality. MOF-regulated genes are highly enriched for essential functions in the mitochondria and cilia. Genetic deletion of *Uqcrq*, an essential subunit for the electron transport chain (ETC) Complex III, in the skin, recapitulates the defects in epidermal differentiation and hair follicle growth observed in MOF knockout mouse. Together, this study reveals the requirement of MOF-mediated epigenetic mechanism for regulating mitochondrial and ciliary gene expression and underscores the important function of the MOF/ETC axis for mammalian skin development.

Histone H4 lysine 16 acetylation (H4K16ac) has been implicated to play important functions, ranging from chromatin decompaction, and gene expression regulation to priming future gene activation[1–6]. Although histone acetylation is generally associated with gene expression activation, H4K16ac is distinct from the acetylation at positions 5, 8, and 12 of histone H4 with specific transcriptional outcomes when ablated in budding yeast[7]. Structural and biophysical studies demonstrate that H4K16ac plays an essential role in transcription activation mediated by its ability to regulate both nucleosome structure and interaction with chromatin-binding proteins, such as epigenetic modifiers for other histone marks[8–10]. H4K16ac is catalyzed by the MYST-family lysine acetyltransferase (KAT) MOF[11,12] (also known as KAT8), which is broadly conserved in flies, mice and humans. However, MOF has been detected in two highly conserved and mutually exclusive complexes, MSL (male-specific lethal) and NSL (non-specific lethal), in fly, mouse, and human

cells[11,13–15]. Whereas MSL is largely specific to H4K16ac[16,17], recent studies in mammalian cells have demonstrated that many functions of MOF, including promoting mtDNA transcription[1], are mediated by NSL, which has a broader range of substrates than H4K16ac[13,16,18]. While these studies have demonstrated complex functions of MOF in mammalian cells[3,19–21], the lack of genetic study for MOF comparatively with the NSL complex has severely hindered the understanding of MOF/ H4K16ac-mediated functions in mammalian development. Furthermore, it remains unclear how the MOF-regulated gene expression program governs cellular functions and lineage differentiation during mammalian tissue development. Mechanistically, although H4K16ac has the unique ability to decompact chromatin structure and orchestrate other chromatin marks to activate gene expression[7,9,10,22], whether specific transcription factors (TFs) require MOF and H4K16ac to regulate gene expression and cellular functions remains unknown.

[1]Department of Pathology, Northwestern University Feinberg School of Medicine, Chicago, IL 60611, USA. [2]Robert H. Lurie Comprehensive Cancer Center, Northwestern University Feinberg School of Medicine, Chicago, IL 60611, USA. [3]Department of Medicine, Northwestern University Feinberg School of Medicine, Chicago, IL 60611, USA. [4]Department of Medicine, University of Southern California, Los Angeles, CA 90033, USA. [5]Department of Dermatology, Northwestern University Feinberg School of Medicine, Chicago, IL 60611, USA. ✉e-mail: yir@northwestern.edu

To address these important questions, we leverage the spatiotemporally well-established lineage specification and tissue morphogenesis in mammalian skin to define the function and mechanism of MOF and H4K16ac. We employed epithelial specific conditional knockout (cKO) mouse models for both *Mof* and *Kansl1*, an essential component of the NSL complex[14,15,23]. Our findings provide genetic evidence that *Mof*, but not *Kansl1*, plays a crucial role in embryonic skin development. Furthermore, we discover that MOF-regulated genes are highly enriched in those involved in two important organelles, mitochondria, and primary cilia, in the skin. Notably, many MOF targets, particularly ciliary genes and, to a lesser extent, mitochondrial genes, are regulated by a family of conserved RFX (Regulatory Factor binding to the X-box) transcription factors (TFs), which preferentially bind to gene promoters and regulate gene expression in a MOF-dependent manner. Finally, we present genetic evidence that MOF-regulated mitochondrial functions, through the electron transport chain (ETC), are critical for epidermal differentiation and hair follicle development.

## Results

### MOF and H4K16ac are required for embryonic skin development

We first examined the functions of MOF during embryonic skin development by conditionally deleting MOF (cKO) with an epithelial specific Cre line (*Krt14-Cre*)[5,24]. Immunofluorescence (IF) staining for H4K16ac, which requires the acetyltransferase activity of MOF[23], in embryonic skin revealed that this histone mark is ubiquitously present in both epithelial and dermal cells of the skin. Homozygous deletion of *Mof* by *Krt14-Cre* (MOF cKO) resulted in a gradual depletion of H4K16ac in the embryonic epidermis (Fig. 1a). At embryonic day 15.5 (E15.5), H4K16ac level was reduced to ~30% of control. By E16.5, the H4K16ac level was further reduced to <20%. In contrast, the H4K16ac signal in the dermis was intact.

Although MOF cKO mice were born at the expected Mendelian ratio and had a similar size to their wildtype (WT) or heterozygous (het) littermate controls, they died within a few hours after birth, indicative of severe defects in the skin. Indeed, most newborn MOF cKOs missed parts of their skin, usually at the flank regions near the forelimbs. To circumvent skin detachment caused by mechanical

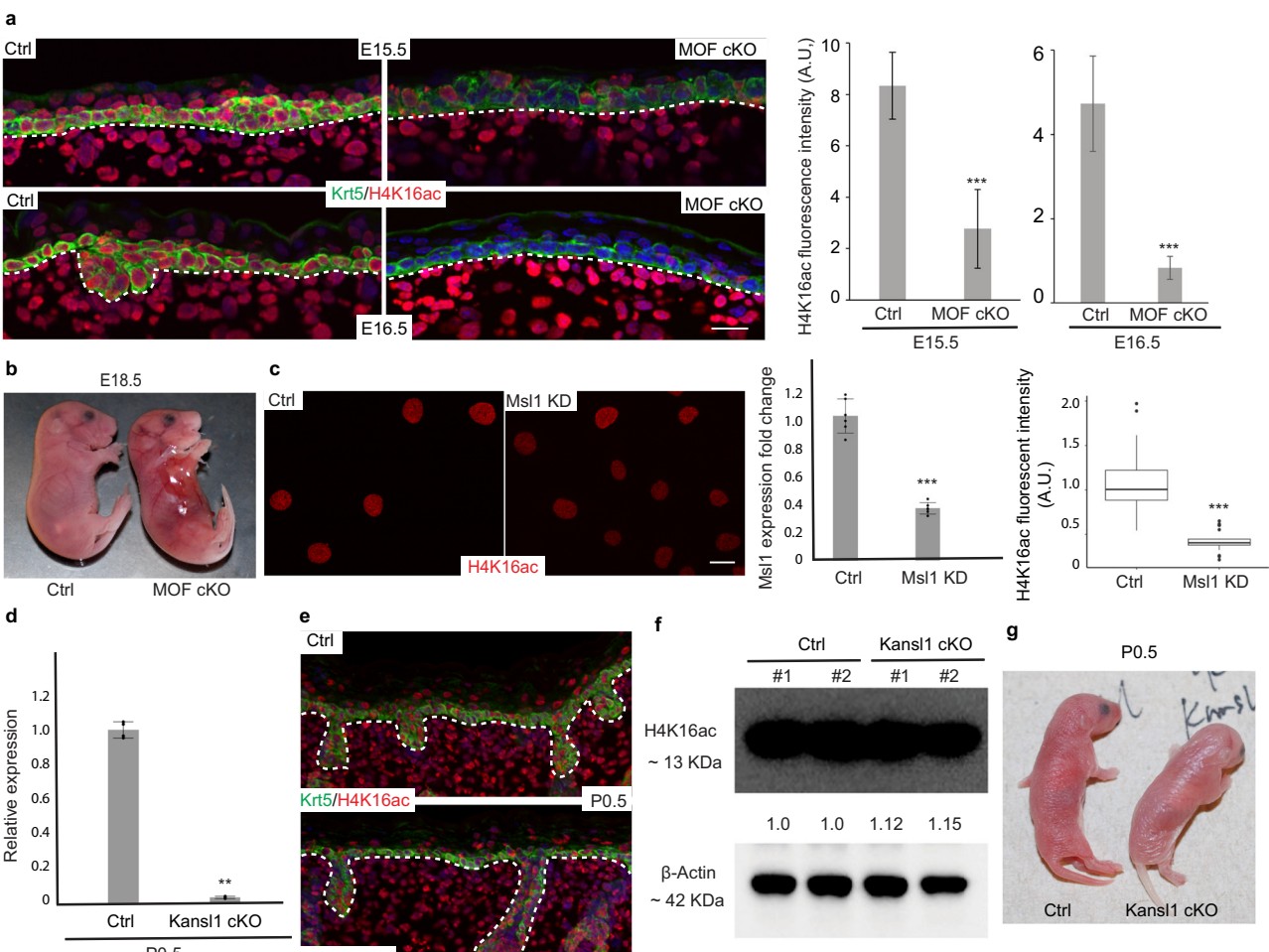

**Fig. 1 | MOF and H4K16ac are required for embryonic skin development.**
**a** Immunofluorescence (IF) staining showing ubiquitous distribution of H4K16ac throughout the embryonic skin in control (Ctrl) animals, and gradual loss of H4K16ac in the epidermis of MOF cKO animals at embryonic day 15.5 (E15.5) and E16.5. And quantification of H4K16ac fluorescence intensity (*n* = 15 cells. Data are represented as mean value ± SEM) for E15.5 and E16.5 samples. **b** Partial loss of skin and thinner skin in MOF cKO animals at E18.5. **c** IF staining and quantification showing reduced H4K16ac upon knockdown of *Msl1* by shRNA in mouse keratinocytes. *n* = 25 cells for control, *n* = 23 cells for *Msl1* KD. Data are represented as

mean value ± SEM. **d** Knockout of *Kansl1* exon3 in epidermis confirmed by qPCR. *n* = 2 animals per genotype. **e** H4K16ac staining in P0.5 control and *Kansl1* cKO. **f** Quantification of H4K16ac level by Western blot in P0.5 *Kansl1* cKO epidermal cells. Two animals per genotype are shown. **g** Indistinguishable appearance of *Kansl1* cKO from control animals at P0.5. Ctrl, control. A.U., Arbitrary Unit. White dashed lines mark the epidermal-dermal boundary (**a**, **e**). *P* values were calculated by unpaired two-sided Student's *t*-test (**a**, **c**, **d**), **P < 0.01; ***P < 0.001. The exact *P* values are shown in Supplementary Data 4. Scale bar, 20 μm. Source data are provided as a Source data file.

stress during birth, we collected embryos at E18.5. Yet, MOF cKO embryos already developed fragile skin that was occasionally detached from the underlying dermis (Fig. 1b). In areas where the skin was present, it was visibly thin, indicating defective epidermal formation. These data provided initial evidence that loss of MOF compromised epidermal adhesion and differentiation during embryonic skin development.

MOF resides in mutually exclusive and evolutionarily conserved MSL and NSL complexes. Although both complexes are capable of acetylating H4K16, MSL has been identified as the main HAT complex acetylating H4K16[16,20,21]. Consistent with this notion, shRNA knockdown of *Msl1*, an essential component of MSL[16,23], resulted in significantly reduced H4K16ac in primary mouse keratinocytes (Fig. 1c). In contrast to the MSL complex, non-histone acetylation activities of MOF, such as acetylation of p53 and LSD1[13,16,18] as well as intramitochondrial gene regulation[1], are restricted to the NSL complex.

To examine whether the severe defects observed in MOF cKO are mediated by deacetylation of H4K16 or other substrates of MOF mediated by NSL, we deleted an essential component of NSL, *Kansl1* also known as MSL1v1[13,15], in epithelial cells of the skin by using the same *Krt14-Cre* (Kansl1 cKO). The knockout strategy removes the third exon of *Kansl1* (Supplementary Fig. 1a), which generates two consecutive premature stop codons immediately in the next exon and abrogates the expression of well-characterized functional domains of KANSL1, including the WDR5 binding domain and MOF binding domain[15]. Epithelial cell-specific deletion was confirmed by genotyping (Supplementary Fig. 1b), qPCR (Fig. 1d), and RNA-seq (Supplementary Fig. 1c). Genetic deletion of *Kansl1* in the epidermis, however, did not strongly affect H4K16ac as shown by IF staining (Fig. 1e) and western blot (Fig. 1f). In sharp contrast to the severe defects observed in MOF cKO mice, *Kansl1* cKO mice were indistinguishable from control littermates (Fig. 1g) and lived normally. Upon closer examination, a slight increase of apoptosis was detected in *Kansl1* cKO epidermis (Supplementary Fig. 1d). Basement membrane (BM) marked by β4 integrin, epidermal differentiation marked by KRT1, hair follicle progenitors marked by SOX9 and cell proliferation marked by Ki67 all appeared normal and indistinguishable between control and *Kansl1* cKO (Supplementary Fig. 1e). Furthermore, RNA-seq of control and *Kansl1* cKO epidermal samples only identified 51 downregulated genes and 35 upregulated genes in *Kansl1* cKO (Supplementary Fig. 1f), consistent with the minimum perturbation to skin development. Taken together, these data establish the requirement of MOF-controlled H4K16ac during embryonic skin development and reveal the dispensable role of the NSL complex for MOF-mediated functions in the skin.

## Loss of MOF compromises basal cell adhesion, epidermal differentiation, and hair follicle morphogenesis

We next performed an in-depth examination of epidermal defects in MOF cKO skin, which tracked closely with H4K16ac depletion. At E15.5, when H4K16ac was reduced by 70%, the MOF cKO epidermis appeared thinner (Fig. 2a). By E16.5 when H4K16ac was largely depleted in the MOF cKO epidermis, epithelial layers of MOF cKO skin started to show signs of reduced adhesion and had fewer differentiated layers (Fig. 2a). By E18.5, epidermal detachment became widespread, and the reduction of basal cells and compromised epidermal differentiation, shown by reduced granular layers and stratum corneum, were evident in morphological analysis (Supplementary Fig. 2a, b). To determine molecular features associated with epidermal detachment, we examined basal epithelial cells for the integrity of the BM. In MOF cKO, β4 integrin, a key component connecting basal epidermal cells to the BM, was detected in KRT5+ basal cells but showed a diffused distribution in the basal cells of E15.5 skin and largely failed to localize to the BM at E16.5 (Fig. 2b). Moreover, collagen XVII, a key component of the BM, largely lost the localization to the BM (Supplementary Fig. 2c). Consistent with these molecular defects, electronic microscopy (EM)

revealed a strong reduction of hemidesmosome and, in some regions, a complete loss of the BM (Fig. 2c), corroborating the disorganized patterns of β4 and collagen XVII. Properly formed BM not only promotes epithelial adhesion but also allows the establishment of basal cell polarity, which, in turn, permits epidermal stratification and differentiation[25]. Consistent with the severely compromised BM and basal cell-BM adhesion, MOF cKO basal cells also failed to establish polarity, revealed by disoriented Pericentrin signals (Fig. 2d) and disorganized adherens junctions (Fig. 2e). Instead of apical and lateral localization of E-Cadherin observed in control basal cells, E-Cadherin mis-localized to the basal side in MOF cKO, reflecting lost polarity and compromised adhesion to the BM. The strongly compromised epithelial cell adhesion and BM prompted us to examine whether epithelial fate specification was altered in the absence of MOF and H4K16ac. However, the robust expression of KRT5 and ΔNp63 (Supplementary Fig. 2d), the master TF required for epidermal fate specification[26], ruled out the possibility of failed epidermal fate in MOF cKO.

We next examined epidermal differentiation. At E15.5 prior to the significant depletion of H4K16ac (Fig. 1a), both KRT5+ basal and KRT1+ spinous layers were largely intact in MOF cKO epidermis although the overall thickness of the epidermis was slightly reduced in cKO (Fig. 2f). However, by E16.5, whereas control epidermis continued to thicken with the expansion of differentiated spinous and granular layers, MOF cKO epidermis had much thinner spinous and granular layers (Fig. 2f, g). Similarly, basal cells were in the active cell cycle, as indicated by universal Ki67 signals, in both control and MOF cKO skin at E15.5 (Supplementary Fig. 3a). By E16.5, however, basal cell proliferation was reduced, and the detached epidermal regions showed the strongest reduction (Supplementary Fig. 3b). Apoptotic cells, marked by activated Caspase 3, were detected in a relatively low frequency in the basal layer of the cKO epidermis at E15.5 (Supplementary Fig. 2c), and the detection frequency was markedly elevated by E16.5 (Supplementary Fig. 2d). Furthermore, reduced cell adhesion to the BM caused premature delamination of basal cells at E16.5. Interestingly, many of these delaminated cells were stalled to a KRT5+/KRT1− basal-like phenotype (Fig. 2f, arrowheads), failing to differentiate. Collectively, defects in cell adhesion and differentiation severely compromised epidermal integrity in the suprabasal layers (Fig. 2f). Consistent with defective epidermal differentiation, granular layer, marked by Loricrin (Lor), was also thinner and had weaker Lor signals in MOF cKO (Fig. 2g). Finally, KRT6 was robustly detected in the suprabasal layers of MOF cKO epidermis at E16.5 (Supplementary Fig. 3e), reflecting a stressed cellular state.

We then turned to hair follicle (HF) induction and morphogenesis. Overall, the number of morphologically discernible HFs was significantly reduced (Fig. 2a). In the few HFs that were still present in cKO, however, the stem cell/progenitor marker, SOX9, was readily detected (Supplementary Fig. 3f). This argued against the complete failure of HF fate specification. Indeed, LEF1, the bona fide TF for the activated WNT signaling pathway and the marker for hair placode formation[27,28] was still detected in regularly spaced hair placodes in MOF cKO skin, where the BM was relatively intact, at E16.5 (Supplementary Fig. 3g). These data suggest that HF fate was induced but the downward growth of HFs was arrested in the absence of MOF and H4K16ac, reminiscent of the hair growth defects observed in Shh KO[29,30]. Altogether, these data illustrate the widespread impact of MOF and H4K16ac on epidermal and HF cell lineages.

## Single-cell RNA-seq reveals widespread downregulation of nuclear encoded mitochondrial genes

We next performed single-cell RNA-seq (scRNAseq) to interrogate the changes of transcriptome and cellular states caused by the deletion of *Mof* at E15.5 when H4K16ac was not completed depleted yet and when epithelial cells showed relatively minor defects. Dorsal skin samples

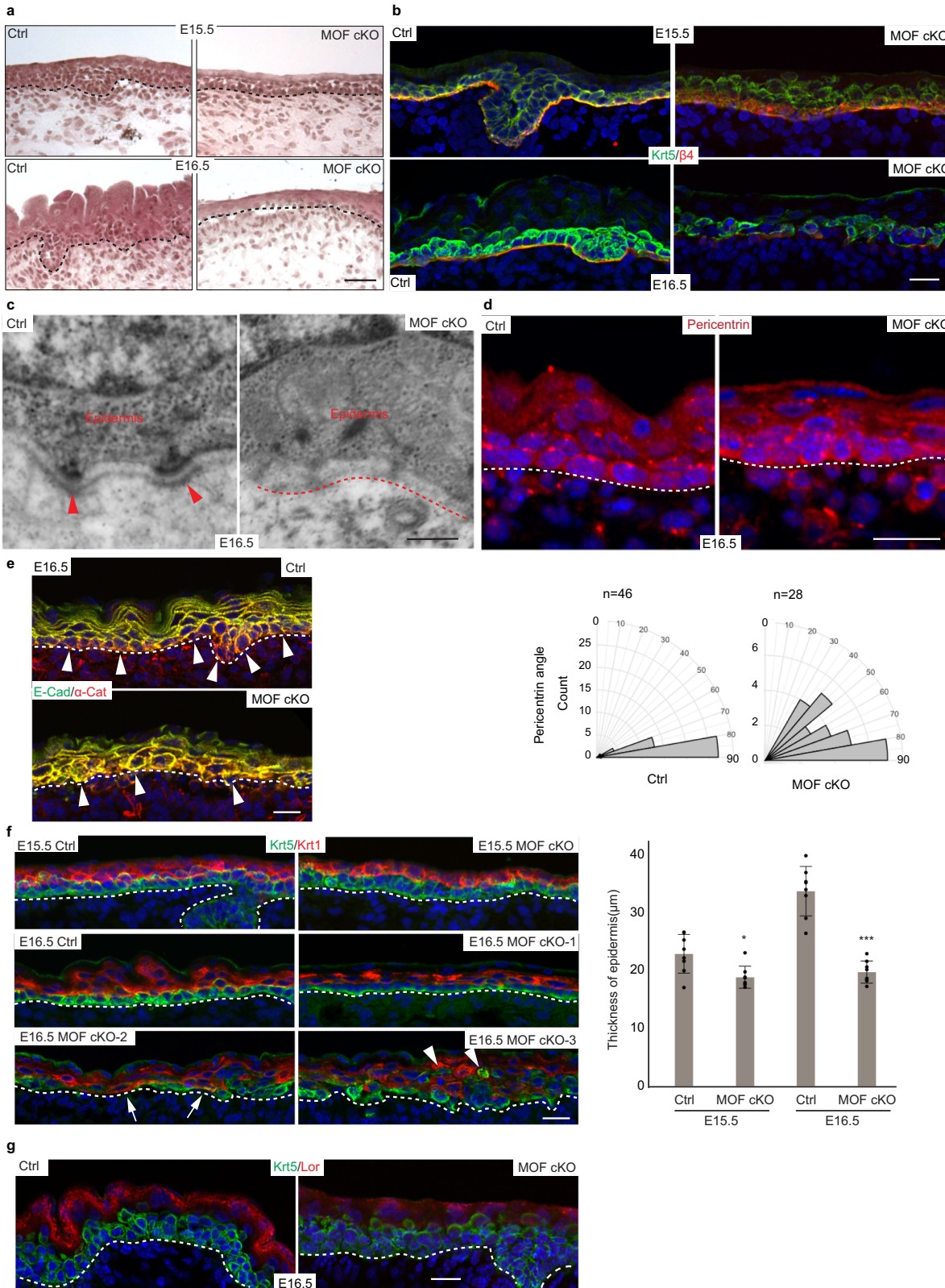

from two pairs of control and MOF cKO embryos were profiled with the 10x Chromium Single Cell 3' kit. After quality control, nine major cell types were identified in control and MOF cKO samples, including epithelial cells, dermal cells, and immune cells among others (Fig. 3a). In two control samples, epithelial cells represented 31% and 32% of the total population, respectively (Supplementary Data 1). In contrast, in two MOF cKO samples, epithelial cells represented 10% and 15% of the

total, respectively. These results were consistent with the reduced epithelial cell populations as observed in our phenotypical analysis.

We first focused on epithelial cell clusters. Three clusters were detected in control samples, including basal epidermal progenitors (marked by high Krt5 and highly proliferative), hair germ (marked by Sox9 and Krt17), and differentiating suprabasal cells of the epidermis (marked by Krt1) (Fig. 3b and Supplementary Fig. 4a–d). Nearly all

**Fig. 2 | Loss of MOF compromises basal cell adhesion, epidermal differentiation, and hair follicle morphogenesis. a** HE staining at E15.5 and E16.5 showing thinner epidermis in MOF cKO. Representative images from five pairs of samples. **b** Progression of compromised basement membrane in MOF cKO as shown by β4 integrin staining at E15.5 and E16.5, respectively. Representative images from five pairs of samples. **c** Loss of hemi-desmosome and basement membrane shown by electronic microscopy. Red arrowheads in Ctrl indicate hemi-desmosome, and red dashed line in MOF cKO indicates missing basement membrane. Representative images from two pairs of samples. **d** Disorientation of basal cells indicated by pericentrin staining. Representative images from three pairs of samples. *n* indicates the number of counted cells. **e** Loss of polarity in basal cells as shown by the distribution of adheres junction markers E-Cadherin (E-Cad) and α-Catenin (α-Cat).

Note the lack of E-Cad signals in the basal side of the basal cells in Ctrl but not in MOF cKO (white arrowheads). Representative images from three pairs of samples. **f** Thinner epidermis in MOF cKO. Arrows indicate Krt1$^+$ cells came in contact with the basement membrane, arrowheads point to suprabasal Krt5$^+$ cells that did not express Krt1. *n* = 8 measurements. Data are represented as mean value ± SEM. **g** Defect in terminal differentiation indicated by granular layer marker Loricin (Lor) staining. Representative images from five pairs of samples. Ctrl, control. Black (**a**) and white (**d**, **e**, **f**, and **g**) dashed lines mark the epidermal-dermal boundary. *P* values were calculated by unpaired two-sided Student's *t*-test (**f**), \**P* < 0.05; \*\*\**P* < 0.001. The exact *P* values are shown in Supplementary Data 4. Scale bar, 50 μm (**a**), 20 μm (**b**, **d**, **e**, **f**, and **g**), 200 nm (**c**). Source data are provided as a Source Data file.

epithelial cells from one MOF cKO sample (cKO5) clustered completely away from control cells, and these cKO cells were identified as basal progenitor-like and suprabasal-like clusters, respectively (Fig. 3b). Interestingly, some cells from the other MOF cKO sample (cKO6) clustered together with cKO5 cells whereas some cells from the same sample clustered together with control cells (Fig. 3b and Supplementary Fig. 4b). This bimodal distribution pattern suggested that cKO6 was mosaic whereas cKO5 was a homogenous MOF cKO. Indeed, IF staining revealed consistently reduced H4K16ac levels in epithelial cells of MOF cKO5. In contrast, MOF cKO6 showed mosaic H4K16ac patterns in the epidermis (Supplementary Fig. 4e), likely due to mosaic expression of Cre. The unexpected mosaic depletion of MOF/H4K16ac and scRNAseq result demonstrates that the loss of MOF and H4K16ac alters epithelial cell states in a cell-intrinsic manner.

We next examined differentially expressed gene categories in basal and suprabasal epithelial populations between control and MOF cKO. Strikingly, downregulated genes in MOF cKO basal cells were highly enriched in Gene ontology (GO) categories of mitochondrion and, to a lesser extent, cell cycle, proteasome, RNA splicing, protein biosynthesis, and DNA repair (Fig. 3c). The same GO categories for downregulated genes were also enriched in suprabasal cells of MOF cKO (Fig. 3d).

To further investigate the changes in MOF cKO transcriptome, we performed scRNAseq at E16.5 when H4K16ac was largely depleted (Fig. 1a) and morphological defects were more pronounced. Ten distinct populations were detected in E16.5 control skin. We observed further reduced epithelial populations in MOF cKO skin (18% and 21% in two control samples, respectively, and 5% and 3% in two MOF cKO samples, respectively) (Supplementary Data 1). Moreover, we noticed perturbed dermal cell clustering in MOF cKO samples, compared to their counterparts in control samples, although MOF was deleted specifically from epithelial cells and H4K16ac was intact in the dermis (Supplementary Fig. 5a, b). GO enrichment analysis for upregulated genes in MOF cKO dermis revealed that altered collagen fibril organization and increased inflammatory response contributed to the different cellular states (Supplementary Fig. 5c), reflecting responses of dermal cells to MOF cKO epithelium. In support, MOF cKO samples showed more immune cell infiltration (8.4% in cKO vs 3.7% in Ctrl), consistent with the increased inflammatory signatures in the dermal cells. Notably, the altered dermal cell states and increased immune cell infiltration were not detected in E15.5 samples. These results suggest that the E15.5 dataset captured the primary and cell intrinsic changes whereas the E16.5 dataset also captured the secondary changes.

Next, we re-clustered epithelial cells in the E16.5 scRNAseq dataset. Four epithelial clusters were detected in control, including basal epidermal progenitor (marked by high Krt14 and Krt5 and high proliferation), spinous layer of the epidermis (marked by Krt1 and Krt10), granular layer of the epidermis (marked by Lor and Ivl) and HF progenitors (marked by Sox9 and Pthlh) (Fig. 3e and Supplementary Fig. 5d, e). Interestingly, only two major clusters, both of which were distinct from control clusters, were detected in MOF cKO (Fig. 3e and

Supplementary Fig. 5d), underscoring the profound changes in the cKO transcriptome. One cKO cluster, which had high levels of Krt14 and Krt5 as well as other basal markers (Supplementary Data 2), represented the basal cell population of cKO (Fig. 3e and Supplementary Fig. 5e). The other cKO cluster, representing differentiated epithelial cells, could be further separated into two sub-clusters based on differential Krt5 and Krt1 expression patterns (Fig. 3e and Supplementary Fig. 5e). One Krt1$^{high}$/Krt5$^{low}$ sub-cluster probably corresponded to the differentiated spinous layer cells. The other Krt1$^{low}$/Krt5$^{high}$ sub-cluster was reminiscent of the KRT5+/KRT1- suprabasal cells seen in morphological studies (Fig. 2f). Thus, scRNAseq captured major cell populations and identified defective epidermal differentiation in control and MOF cKO skin.

We next performed differential gene expression analysis of basal and suprabasal epithelial populations. In addition to reduced gene expression in BM and ECM (Supplementary Fig. 5f), which was consistent with morphological analysis, downregulated genes in MOF cKO basal cells were highly enriched in GO categories of mitochondrion, cell cycle, mRNA splicing, and DNA repair (Fig. 3f). The same GO categories for downregulated genes were also enriched in MOF cKO vs control suprabasal cells (Supplementary Fig. 5g). In addition, genes associated with cornified envelope were the most significantly downregulated in MOF cKO suprabasal cells, confirming the strongly compromised epidermal differentiation (Supplementary Fig. 5g).

To confirm scRNAseq data, we performed in situ hybridization for two genes, Scp2, a peroxisomal lipid carrier gene[31], and Opa1, a nuclear-encoded mitochondrial gene[32], and validated their downregulation in MOF cKO epidermis at E15.5 (Fig. 4a). In addition, shRNA knockdown of *Msl1* in cultured mouse keratinocytes also downregulated *Abca11, Opa1,* and *Scp2*, further supporting the requirement of MSL for MOF-mediated gene expression in the skin (Fig. 4b).

In both E15.5 and E16.5 scRNAseq data, mitochondrial genes were the top enriched category in downregulated genes. We next examined the expression of nuclear-encoded mitochondrial genes in basal and suprabasal cells in control and MOF cKO. By using a mitochondrial gene expression score, which was calculated by using a curated list of 1158 mitochondrial genes, we observed that their expression was elevated in differentiated suprabasal cells, compared with basal progenitors, in control. Furthermore, in both basal and suprabasal cells, the mitochondrial gene expression score was strongly downregulated in MOF cKO (Fig. 4c and Supplementary Fig. 6a). Strikingly, downregulated mitochondrial genes function across different compartments of the mitochondria (Supplementary Fig. 6b). We noticed that many essential components of the electron transport chain (ETC) Complexes I, II, III, IV, and ATP synthase complex were significantly downregulated in both basal and suprabasal MOF cKO epithelial cells at E15.5 (Fig. 4d). Many of these ETC genes were also downregulated in MOF cKO epithelial cells at E16.5 (Fig. 4e). Collectively, these data reveal strong changes of transcriptome and cellular states with nuclear-encoded mitochondrial genes being the most significantly affected category, caused by loss of MOF, during epidermal development.

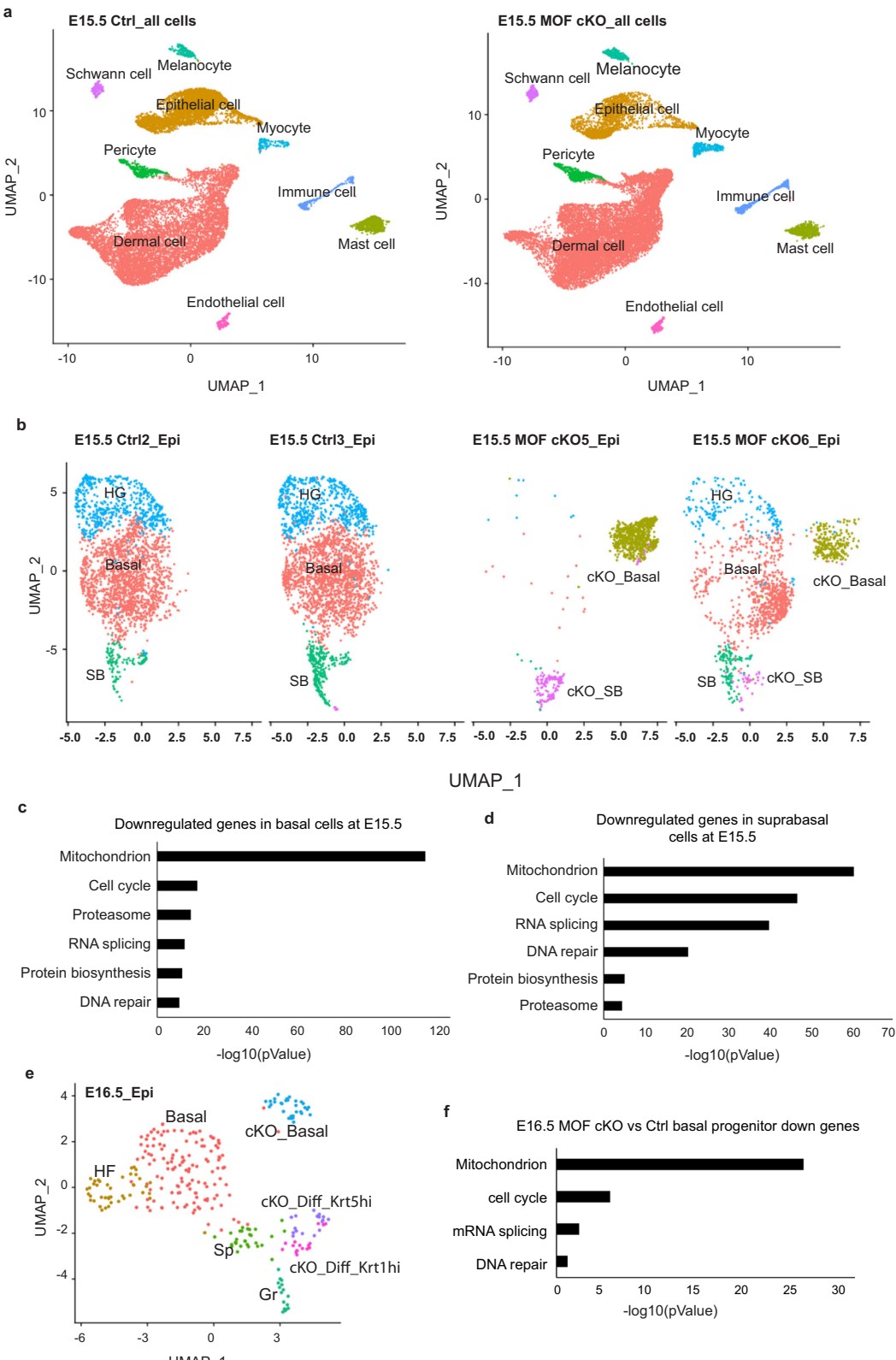

**Fig. 3 | Single-cell RNAseq reveals widespread downregulation of nuclear encoded mitochondrial genes. a** Nine distinct cell types were detected in E15.5 control and MOF cKO dorsal skin samples by scRNAseq. **b** UMAP clustering of epithelial cells from E15.5 scRNAseq, split by sample. **c** Enriched gene ontology (GO) terms of downregulated genes in E15.5 MOF cKO *vs.* control basal progenitor comparison. **d** Enriched GO terms of downregulated genes in E15.5 MOF cKO *vs.* control suprabasal comparison. **e** UMAP clustering of epithelial cells for

E16.5 scRNAseq. **f** Enriched GO terms of downregulated genes in E16.5 MOF cKO *vs.* control basal progenitor comparison. Ctrl, control; Epi, epithelial cells; Basal, basal progenitor cells; HG, hair germ cells; HF hair follicle cells; SB, suprabasal layer cells; Sp, spinous layer cells; Gr, granular layer cells; cKO_Basal, basal like progenitor cells in MOF cKO; cKO_SB, suprabasal like cells in MOF cKO; cKO_Diff_Krt5$^{hi}$, Krt5$^{high}$Krt1$^{low}$ suprabasal cells in MOF cKO; cKO_Diff_Krt1$^{hi}$, Krt5$^{low}$Krt1$^{high}$ suprabasal cells in MOF cKO.

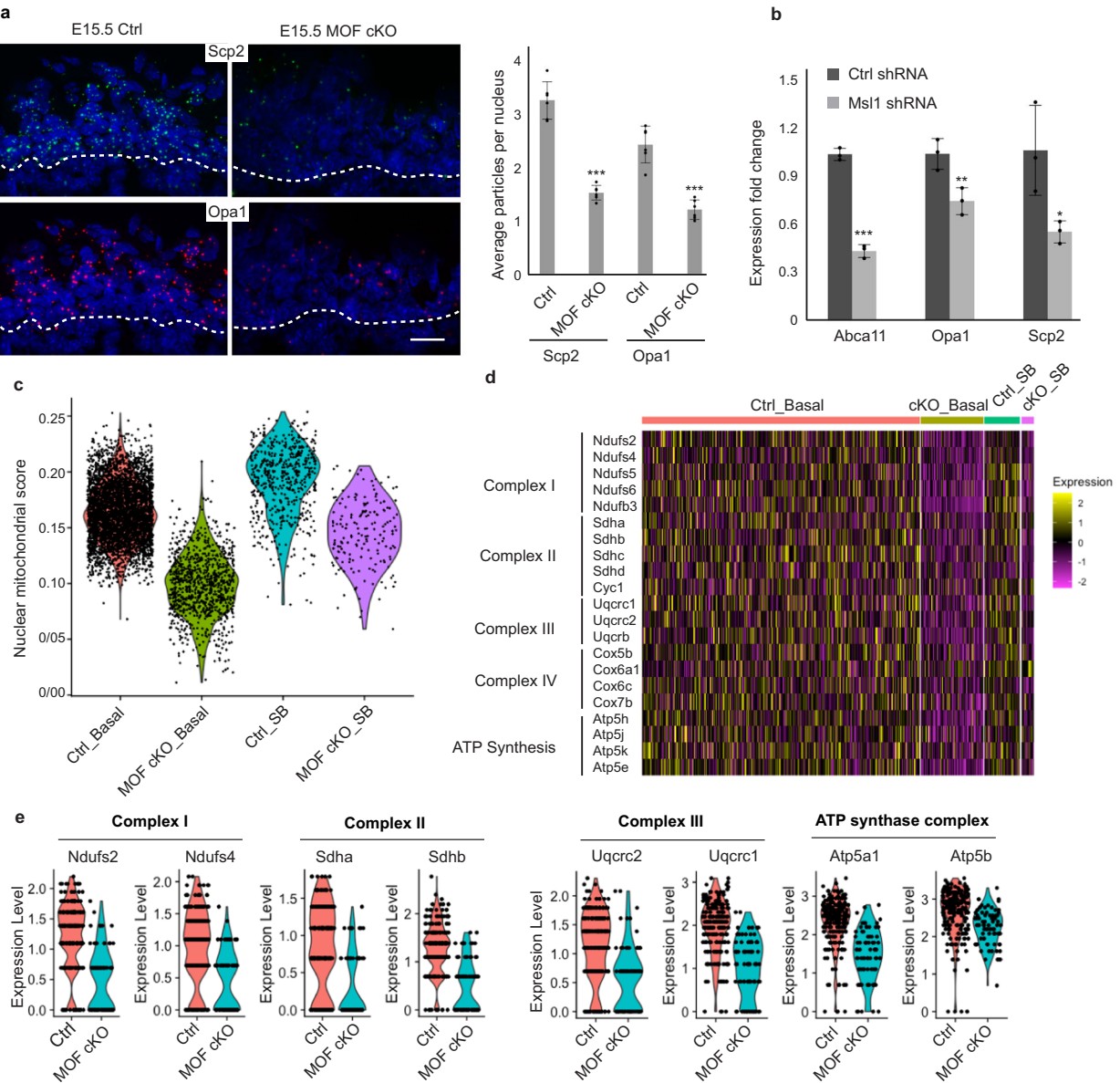

**Fig. 4 | Downregulation of nuclear-encoded mitochondrial genes in MOF cKO.**
**a** In situ hybridization confirmed downregulation of two genes, *Scp2* and *Opa1*, in MOF cKO. White dashed lines mark the epidermal-dermal boundary. Scale bar, 20 μm. *n* = 6 fields from two pairs of samples. Data are represented as mean value ± SEM. **b** qPCR validation of downregulated mitochondrial genes in *Msl1* KD mouse keratinocytes. *n* = 3 replicates. Data are represented as mean value ± SEM. **c** Aggregated nuclear-encoded mitochondrial gene expression score in

E15.5 scRNAseq. **d** Heatmap of downregulated ETC complex genes in E15.5 scRNAseq. **e** Violin plot of downregulated ETC complex genes in E16.5 scRNAseq. Ctrl, control; Ctrl_Basal, control basal cells; cKO_Basal, MOF cKO basal cells; Ctrl_SB, control suprabasal cells; cKO_SB, MOF cKO suprabasal cells. *P* values were calculated by unpaired two-sided Student's *t*-test (**a**, **b**), *$P < 0.05$; **$P < 0.01$; ***$P < 0.001$. The exact *P* values are shown in Supplementary Data 4. Source data are provided as a Source Data file.

## Genetic deletion of QPC recapitulates skin defects of MOF cKO

Having demonstrated that the MOF-regulated gene expression program is important for mitochondria, we next sought to determine how compromised mitochondrial functions cause the defects in MOF cKO. We noticed that a significant number of ETC complex genes were downregulated in MOF cKO. These results prompted us to examine whether defects in MOF cKO were mediated, in part, by compromised ETC. Because the genetic deletion of ETC complex III component *Uqcrq* (QPC) disrupts the entire ETC[33], we used *Krt14-Cre* to delete *Uqcrq* (QPC) and examined the phenotype. Strikingly, QPC cKO mice died within hours after birth, mimicking MOF cKO. Although QPC cKO mice were similar in size compared to WT or het control, their skin was thin and transparent, indicative of epidermal defects (Fig. 5a). Morphological analysis confirmed the thin epidermis and stunted HF

growth such that many HFs failed to grow downward into the dermis (Fig. 5b and Supplementary Fig. 7a). IF signals of Krt1, a spinous layer marker, and Lor, a granular layer marker, showed significantly reduced thickness of the epidermis and compromised epidermal differentiation (Fig. 5c). Distinct from MOF cKO skin, however, the BM was intact with the proper signals of β4 integrin (Supplementary Fig. 7b) and epidermal proliferation showed no change at E18.5 (Supplementary Fig. 7c). These data suggest that the epithelial defects of QPC cKO skin were manifested in epidermal differentiation and HF growth.

To compare defective gene expression caused by the deletion of QPC to that of MOF cKO, we performed scRNA-seq of QPC cKO skin (Supplementary Fig. 7d, e). Overall, the basal epithelial cell population, which self-renews and gives rise to differentiated suprabasal cells, was reduced in QPC cKO (Fig. 5d, e). In comparison, the basal cell

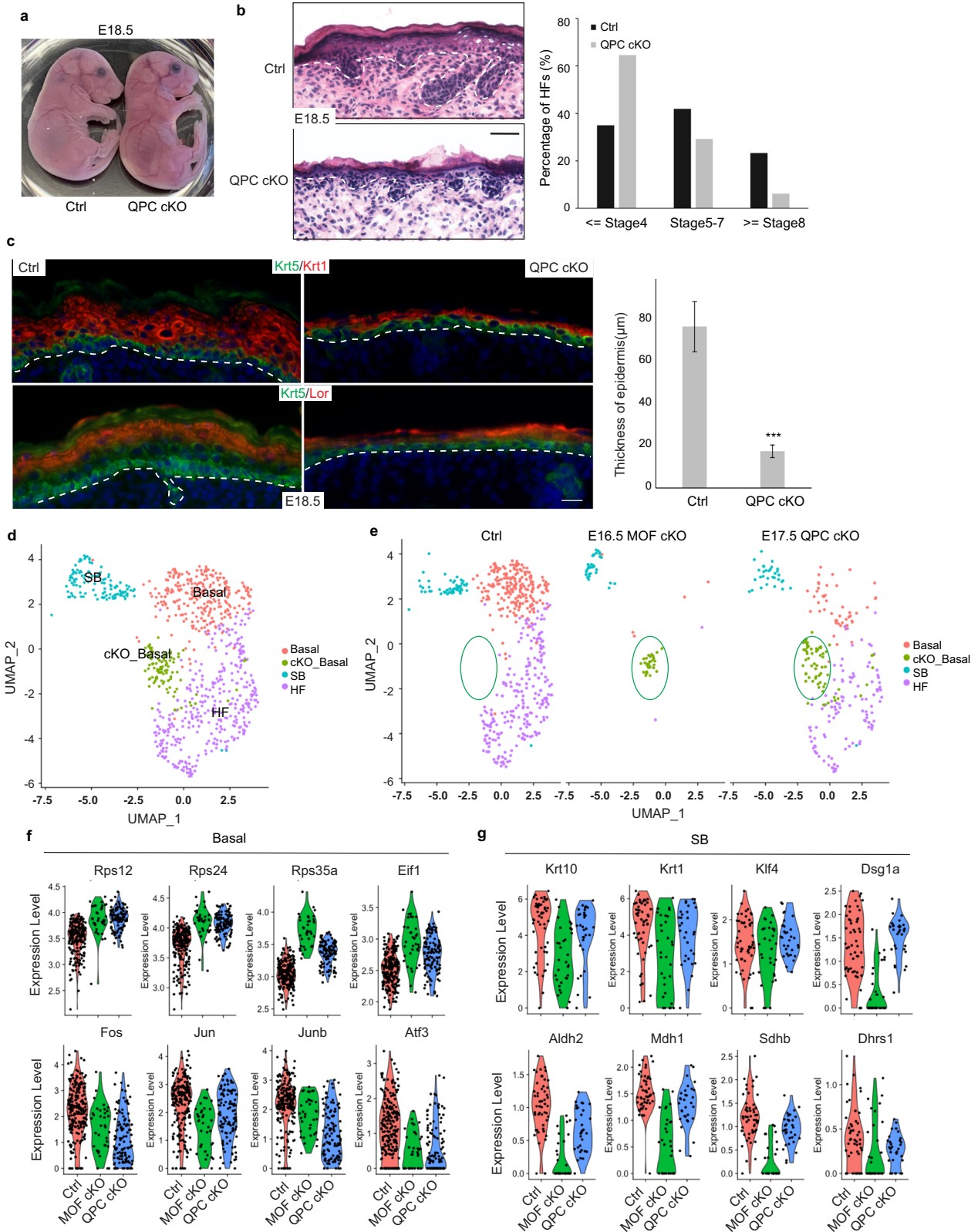

population was largely depleted in MOF cKO (Fig. 5e). Interestingly, a unique basal epithelial cell population was detected in both MOF and QPC cKO but not in control skin (Fig. 5e), indicating similar changes of transcriptome and cellular state in the basal cells of MOF and QPC cKO. Closer inspection revealed that this basal epithelial cell population had highly elevated gene expression in ribosomal genes and translation (Fig. 5f), revealing a similar response of epidermal cells to the deletion

of MOF and QPC. In addition, the basal cells in both MOF and QPC cKO had reduced AP1 TF expression, including *Fos*, *Jun*, *Junb*, and *Atf3* (Fig. 5f), which play important roles in epidermal differentiation[26,34,35]. In suprabasal cells, the commonly downregulated genes were enriched in differentiation and keratin genes, such as *Krt10*, *Krt1*, *Klf4*, and *Dsg1a*, and oxidoreductase activity, such as *Aldh2*, *Mdh1*, *Sdhb* and *Dhrs1* (Fig. 5g). Interestingly, the Krt1$^{low}$/Krt5$^{high}$ population in the

**Fig. 5 | Genetic deletion of QPC recapitulates skin defects of MOF cKO.**
**a** Transparent appearance of QPC cKO, compared with control (Ctrl) mice, at E18.5.
**b** HE staining showing thinner epidermis and lacking advanced stage hair follicles (HF) in QPC cKO. Representative images from three pairs of samples.
**c** Differentiation defects and thinner epidermis in QPC cKO indicated by Krt1 and Loricrin (Lor) staining. $n = 20$ measurements from three pairs of samples. Data are represented as mean value ± SEM. $P$ value was calculated by unpaired two-sided Student's $t$-test, ***$P < 0.001$. The exact $P$ value is shown in Supplementary Data 4.
**d** Integration and clustering of E16.5 control and MOF cKO epithelial cell with E17.5 control and QPC cKO epithelial cell scRNAseq. **e** Split view of Ctrl, MOF cKO, and QPC cKO epithelial cells, as clustered in (**d**). **f** Examples of commonly upregulated genes, related to protein translation, and commonly downregulated AP1 transcription factor genes, in MOF cKO and QPC cKO basal progenitor cells. **g** Examples of commonly downregulated, differentiation related genes and oxidoreductase activity genes in MOF cKO and QPC cKO suprabasal cells. White dashed lines mark the epidermal-dermal boundary (**b**, **c**). Ctrl, control; Basal, basal progenitor cells; HF, hair follicle cells; SB, suprabasal layer cells; cKO_Basal, MOF and QPC cKO basal-like cells. Scale bar, 50 μm (**b**), 20 μm (**c**). Source data are provided as a Source Data file.

suprabasal cluster was only highly abundant in MOF cKO but not in control or QPC cKO (Supplementary Fig. 7f). This observation suggested that those Krt1$^{low}$/Krt5$^{high}$ cells prematurely detached from the basal layer and failed to embark on epidermal differentiation process in MOF cKO. In contrast, QPC cKO cells only showed compromised epidermal differentiation but not adhesion to the intact BM. Together, these data provide genetic and molecular evidence that MOF-controlled epidermal differentiation is mediated, at least in part, by its regulation of the ETC.

### MOF cKO causes widespread downregulation of ciliary genes

MOF and H4K16ac have been implicated in the regulation of "housekeeping" genes in *Drosophila*[36,37], which are generally defined by their relatively high expression and essential functions across different cell types. In this regard, genes associated with mitochondrial functions, cell cycle, mRNA splicing, and DNA repair, which were detected as top downregulated gene groups in our scRNAseq analysis, appeared to corroborate this notion. However, scRNAseq is limited to detecting highly expressed transcripts. To comprehensively identify all genes that are regulated by MOF, we performed bulk RNAseq by using FACS purified epithelial cells isolated from control and MOF cKO skin samples at E16.5 (Supplementary Fig. 8a and 8b). Using FDR ≤ 0.01 and 1.5x fold change as cut-off, we identified 1734 downregulated genes and 1752 upregulated genes (Fig. 6a). Consistent with scRNAseq data, important BM and ECM genes, such as *Col7a1*, *Frem2*, and *Lamc1*, were significantly downregulated (Supplementary Fig. 8c). Furthermore, nuclear-encoded mitochondrial genes, cell cycle and DNA repair genes were also significantly downregulated, mirroring the findings from scRNAseq. Surprisingly, genes associated with primary cilia emerged as the most highly enriched GO category among downregulated genes (Fig. 6b). To further examine the widespread downregulation of ciliary genes, we curated a list of 659 ciliary genes from the GSEA[38], among which 443 genes were detected in the bulk RNAseq, and performed GSEA analysis. The result confirmed the global downregulation of ciliary genes in epithelial cells of MOF cKO (Fig. 6c). Upon closer inspection, these downregulated genes were broadly involved in both biogenesis and function of primary cilia[39], including ciliary basal body, transition zone, intraflagellar transport and Shh signaling (Fig. 6d). Most of these ciliary genes were not detectable in either control or cKO scRNAseq datasets because of their relatively low expression. However, for a few that can be detected in both basal epidermis and hair germ in E15.5 scRNAseq, they were significantly reduced in MOF cKO (Fig. 6e). Hedgehog signaling is mediated through primary cilia[39] and is involved in HF down growth[29]. Similar to their downregulation in bulk RNAseq, *Shh*, and targets, *Gli1*, *Ptch1*, and *Ptch2* were also reduced in MOF cKO in E15.5 scRNAseq (Supplementary Fig. 8d).

These molecular findings prompted us to examine primary cilia in MOF cKO skin. In control, primary cilia were robustly detected in both basal progenitors and hair germs with the polarized localization, consistent with previous reports[40]. In contrast, primary cilia were strongly reduced in their numbers in both basal progenitors and hair germs in MOF cKO (Fig. 6f, g). Together, these data reveal that numerous ciliary genes were downregulated in MOF cKO and uncover the requirement of MOF-regulated transcriptome for the formation of primary cilia in the skin.

### MOF regulates gene expression preferentially through promoters

We next investigated the mechanism of MOF-mediated regulation. We performed MOF ChIPseq and Cut&Run assays for H4K16ac, H3K4me3, H3K4me1, and H3K27ac by using freshly isolated epidermal keratinocytes from newborn animals. Consistent with previous studies, most peaks (3100 peaks from the c1/c2/c3 clusters) co-occupied by MOF and H4K16ac were generally associated with active promoters and enhancers (Fig. 7a), illustrated by the loci of *Mof* and *Ndufs2* (Fig. 7b). We also detected 1799 peaks from the c4 cluster, which were occupied by MOF, H4K16ac, H3K4me1, and H3K27ac but not H3K4me3. In addition, we also detected 5915 peaks from the c5/c6 clusters that were occupied only by MOF but not H4K16ac or other marks, probably due to technical issues related to MOF ChIP and Cut&Run.

Because MOF has been shown to interact with MLL1, a H3K4 methyltransferase, and the interaction leads to coordination between H4K16ac and the methylation marks of H3K4[2], we stained for H3K4me3 and H3K27ac, which mark active promoters[41] and active enhancers[42,43], respectively, in control and MOF cKO skin. Interestingly, H3K4me3 signals were specifically downregulated in MOF cKO epithelial cells but not in adjacent dermal cells (Fig. 7c). In contrast, H3K27ac signals were unchanged in both epithelial and dermal cells of MOF cKO (Fig. 7c). This result suggests that MOF and H4K16ac globally coordinate with H3K4me3 in epithelial cells, as one may expect.

Because genes that were downregulated upon MOF deletion showed a strong enrichment for MOF binding on their promoters, we next focused on these functional targets of MOF, which were defined as genes not only bound by MOF at their promoters but also downregulated in cKO as measured by bulk RNAseq (because it detected more lowly expressed genes than scRNAseq) (Fig. 7d). Indeed, genes associated with centrosome, DNA repair, mitochondrion, cell cycle, and cilium are bona fide MOF direct targets (Fig. 7e).

To determine if any TFs coordinate with MOF and H4K16ac at gene promoters, we performed TF motif search at the promoter of downregulated genes in MOF cKO. Interestingly, RFX2 (Regulatory Factor binding to the X-box) and RFX3 motifs were identified as the most highly enriched motifs (Fig. 7f). Notably, RFX factors belong to an evolutionarily conserved family of TFs and play important roles in governing ciliary gene expression in *C. elegans*, mouse, and human[44–46]. We next performed RFX2 Cut&Run to determine RFX2 footprint globally in epithelial cells of the skin (Supplementary Fig. 9a). Interestingly, 79% of RFX2 bound regions (266 peaks) were on gene promoters (Fig. 7g). Among them, 48% (157 genes) of RFX2 bound promoters, including many ciliary genes such as *Ift74* and *Wdr34* (Supplementary Fig. 7b), overlapped with MOF bound promoters (Fig. 7h). We noticed that the number of RFX2 bound gene promoters (328) were higher than the number of RFX2 bound peaks (266). This was caused by the head-to-head configuration of some RFX2 and MOF bound promoters in the genome, many of which are ciliary and mitochondrial genes and downregulated in MOF cKO (Supplementary Fig. 9c, d). GO term analysis of these 157 genes revealed that cilium, DNA repair, cell cycle, and mitochondrion were the most highly enriched gene categories co-regulated by RFX2 and MOF at the promoter (Fig. 7i). In contrast to the strong

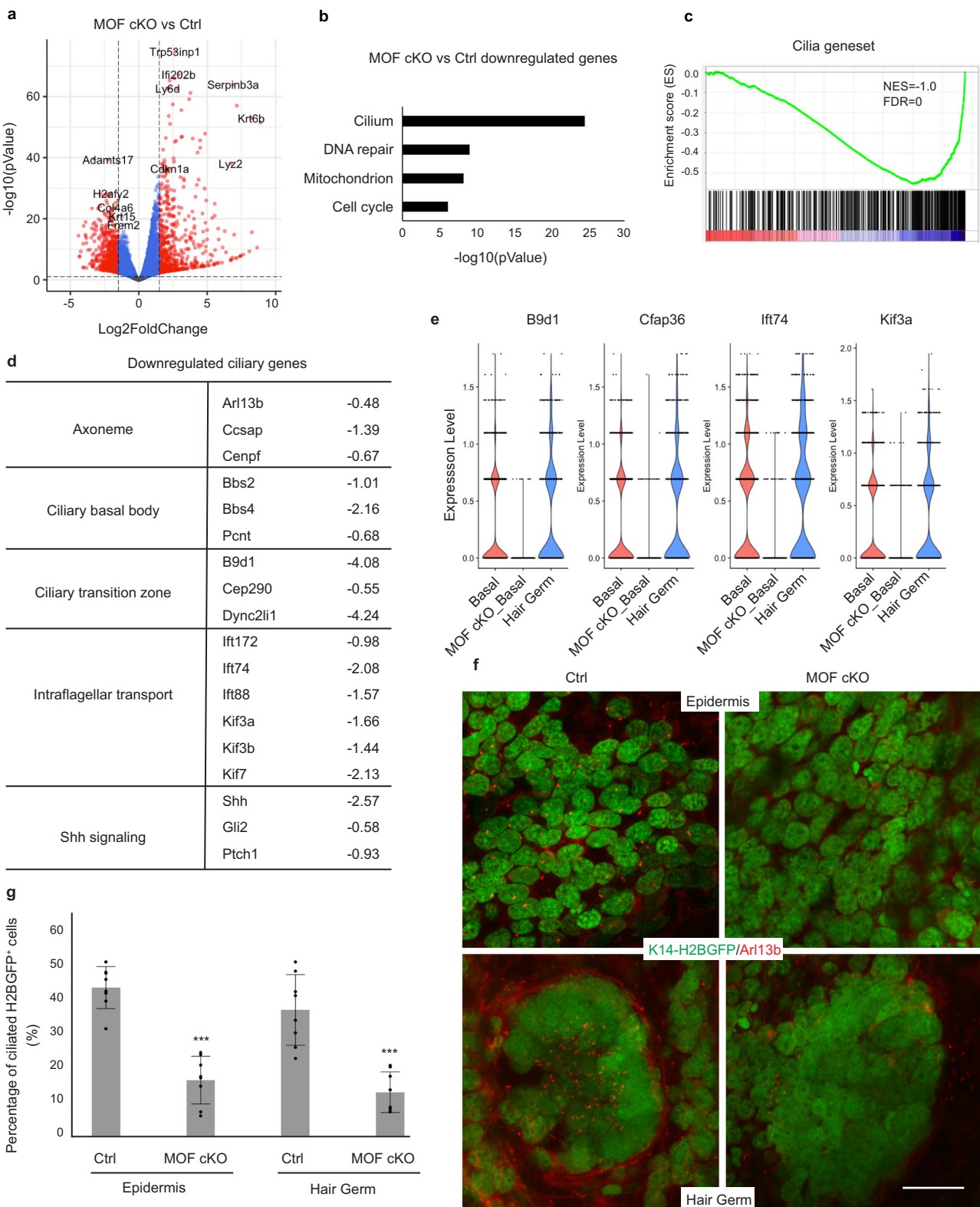

**Fig. 6 | MOF cKO causes widespread downregulation of ciliary genes. a** Volcano plot of differentially expressed genes in E16.5 MOF cKO *vs.* control comparison determined by bulk RNAseq. Red dots represent genes with absolute fold change >1.5 and FDR < 0.01. **b** Enriched GO terms of downregulated genes. **c** GSEA for a set of 443 ciliary genes. **d**, Examples of downregulated ciliary genes in different structures of primary cilia. **e**, Example of downregulated ciliary genes detected in E15.5 scRNAseq. **f** Reduced ciliated epithelial cells in MOF cKO epidermal basal progenitor cells as well as in hair germs as marked by Arl13b. **g** Quantification of the percentage of ciliated K14-H2BGFP⁺ epithelial cells in epidermis and hair germ, respectively. *n* = 8 fields from two pairs of samples. Data are represented as mean value ± SEM. *P* value was calculated by unpaired two-sided Student's *t*-test, \*\*\**P* < 0.001. The exact *P* values are shown in Supplementary Data 4. Scale bar, 20 μm. Source data are provided as a Source Data file.

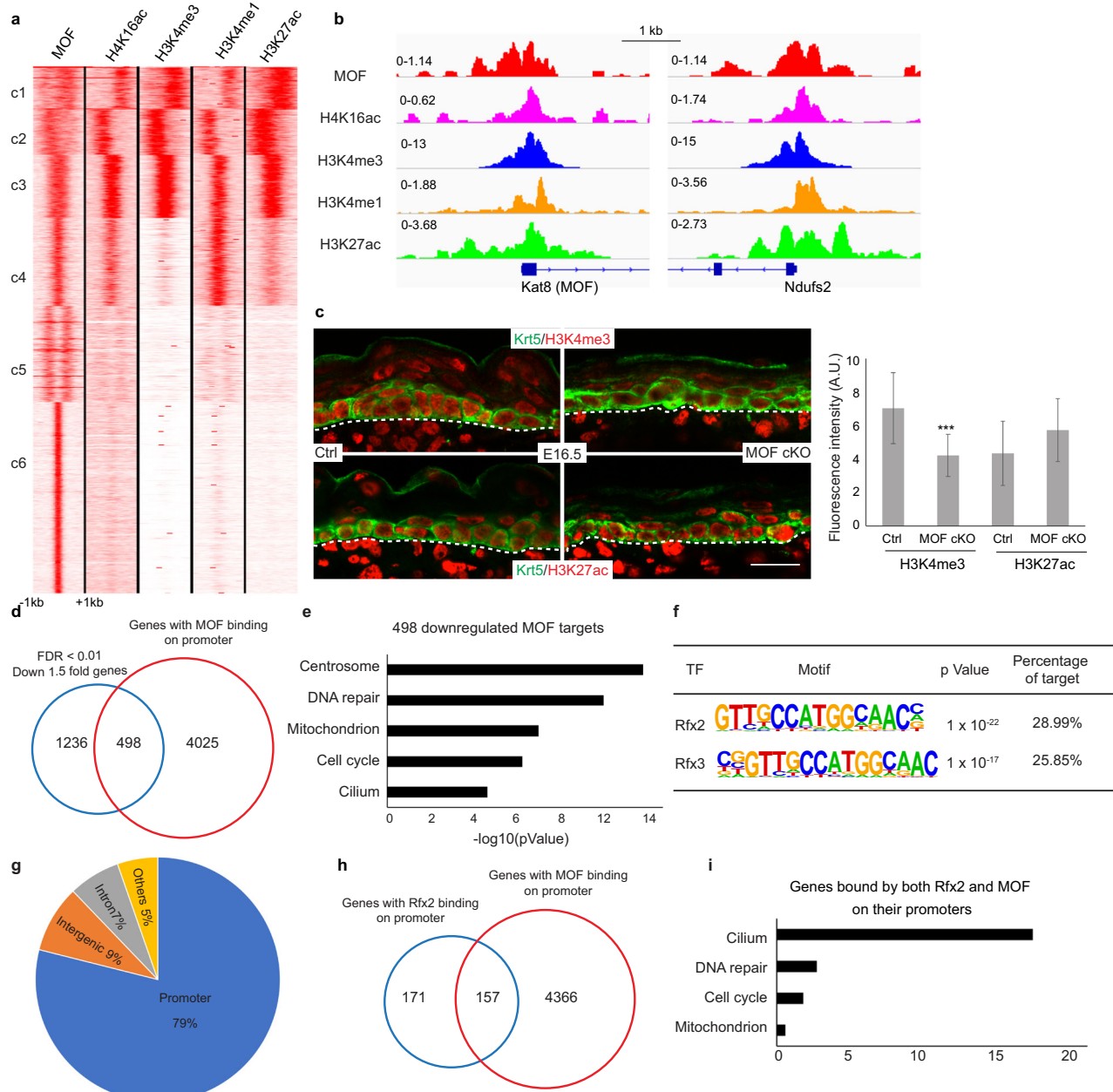

**Fig. 7 | MOF regulates gene expression preferentially through promoters. a** *K*-means clustering of MOF ChIPseq, H4K16ac, H3K4me3, H3K4me1 and H3K27ac Cut&Run data for newborn mouse epidermal keratinocytes. MOF ChIPseq peaks were used as genome coordinates, ±1 kb from the peak center. **b** Integrative Genomics Viewer (IGV) browser view for two gene promoter regions: *Kat8* (MOF) and *Ndufs2*. **c** Reduced promoter mark H3K4me3 and non-changed enhancer mark H3K27ac in MOF cKO. A.U., Arbitrary Unit. White dashed lines mark an epidermal-dermal boundary. *n* = 25 cells for H3K4me3, *n* = 22 cells for H3K27ac from two pairs of samples. Data are represented as mean value ± SEM. *P* value was calculated by unpaired two-sided Student's *t*-test, ***, *P* < 0.001. The exact *P* value is shown in Supplementary Data 4. Scale bar, 20 μm. **d** Venn diagram of downregulated genes in MOF cKO bulk RNAseq with genes that have MOF binding on their promoters. **e** Enriched GO terms for genes that are not only downregulated but also have MOF binding on their promoters, which are defined as MOF targets. **f** Highly enriched motifs on downregulated ciliary gene promoters in MOF cKO. **g** Genomic distribution of RFX2 Cut&Run peaks. **h** Venn diagram of genes with RFX2 binding on their promoters and genes with MOF binding on their promoters. **i** Enriched GO terms for genes with both RFX2 and MOF binding on their promoters. Source data are provided as a Source Data file.

preference to promoters by RFX2, only 16% of regions bound by ΔNp63, a master TF of epithelial cell fate specification[26], were on gene promoters whereas 80% of ΔNp63 bound regions were located in introns or intergenic regions (Supplementary Fig. 9e). Furthermore, there was no enrichment for ΔNp63 regulated targets in either upregulated or downregulated genes in MOF cKO (Supplementary Fig. 9f). Taken together, these results uncover the strong functional association and specificity of MOF and transcription factor RFX2 at gene promoters.

## MOF is required for maintaining chromatin accessibility at promoters

Because H4K16ac plays a critical role in chromatin decompaction[8,22], we next performed scATAC-seq in control and MOF cKO cells isolated from E16.5 skin and determined the changes of chromatin accessibility at promoters and enhancers responding to the loss of MOF and H4K16ac. To enrich epithelial cells, we sorted K14-H2BGFP+ epithelial cells and GFP- non-epithelial cells and mixed them at a 1:1 ratio. We recovered a total of 2132 cells in a control sample, including 1235

epithelial cells, and a total of 4369 cells in MOF cKO sample, including 2,396 epithelial cells. Overall, MOF deletion strongly altered the open chromatin landscape of the epithelial cells, slightly perturbed the dermal fibroblasts but largely spared other cell types in the skin, judging by the clustering patterns (Supplementary Fig. 10a, b).

We next examined the open chromatin status of the transcription start site (TSS). In epithelial cell clusters, the TSS enrichment score, calculated by ArchR[47], was significantly reduced in MOF cKO (Fig. 8a), consistent with globally reduced H3K4me3 signals. As control, the TSS enrichment score remained unchanged for the dermal cells (Supplementary Fig. 10c). Importantly, the normalized insertion score, quantifying the probability of Tn5 insertion and the openness of chromatin[47], was significantly reduced for MOF targeted promoters but not at epithelial lineage gene promoters that are not bound by MOF (see Supplementary Data 3) or at the promoters of dermal cells (Fig. 8b, c and Supplementary Fig. 10d). These data suggest that MOF is required to maintain the open chromatin of its targeted promoters. Indeed, the open chromatin signatures were reduced at the promoter of two MOF-targeted ETC genes, Uqcrc1 and Sdha (Fig. 8d), but showed no

difference at the promoter of Stfa1 (Fig. 8e), which is not a target of MOF. We also noted that the normalized insertion score was significantly higher for MOF targeted promoters than that of the epithelial lineage genes (compare Fig. 8b and 8c). Collectively, these data demonstrate a highly opened chromatin state of MOF-bound promoters and provide experimental evidence for the requirement of MOF and H4K16ac for maintaining chromatin accessibility at targeted promoters.

## Discussion

In this study, we used genetic mouse models and genomic tools to investigate the function and mechanism of MOF and H4K16ac in embryonic skin development. Leveraging spatiotemporally well-defined phenotypical and genomic analysis, we have uncovered specific functions of MOF-regulated transcriptome in governing epithelial cell adhesion, cell cycle, epidermal differentiation, and HF growth (Fig. 8f). Surprisingly, genetic deletion of Kansl1, an essential component of the NSL complex, leaves H4K16ac largely intact and causes minimal disruption of embryonic skin development, in sharp contrast

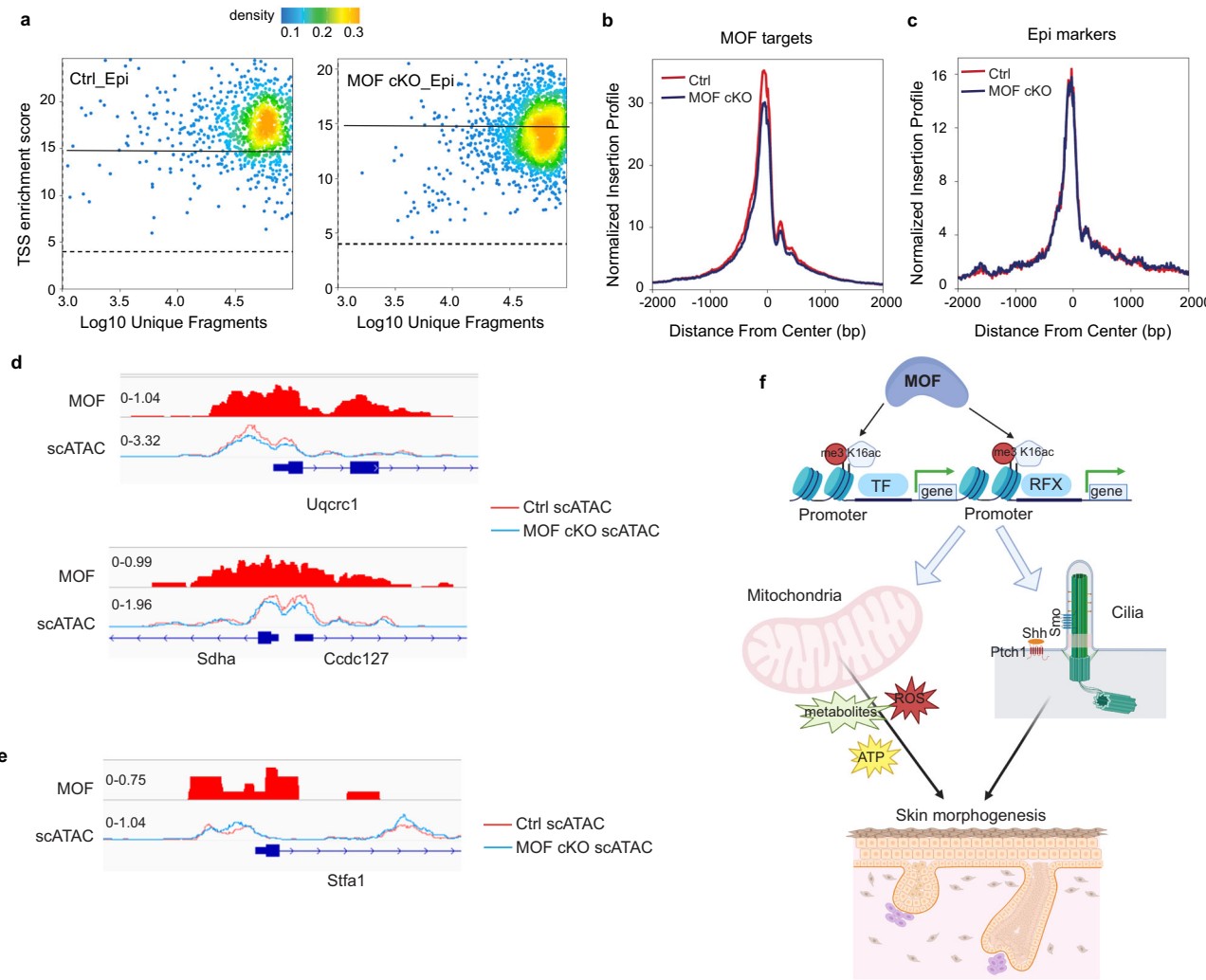

**Fig. 8 | MOF is required for maintaining chromatin accessibility at gene promoters. a** Reduced transcription start site (TSS) enrichment score (y axis) in MOF cKO epithelial cells when compared with those of control cells. Ctrl_Epi, control epithelial cells, MOF cKO_Epi, MOF cKO epithelial cells. **b** Reduced normalized Tn5 insertion profile calculated from scATACseq data for MOF target genes in MOF cKO. **c** Unchanged normalized Tn5 insertion profile calculated from scATACseq data for epithelial marker genes, which are not targeted by MOF. **d** Integrative

Genomics Viewer (IGV) browser view for two mitochondrial genes showing MOF binding on their promoters and reduced promoter openness in MOF cKO epithelial cells determined by scATACseq. **e** IGV browser view of Sfta1 with non-changed promoter openness in MOF cKO epithelial cells determined by scATACseq. **f** Schematic illustration of MOF/H4K16ac-regulated gene expression controlling mouse skin morphogenesis through regulating mitochondria and primary cilia. Created with BioRender.com.

to MOF cKO. Because previous studies have demonstrated that NSL has a broader substrate specificity and mediates the acetylation of non-histone proteins such as p53 and LSD1 as well as intramitochondrial functions of MOF[1,13,15,18], these results strongly argue that histone acetyltransferase activities of MOF are essential for embryonic skin development. Furthermore, biochemical studies have demonstrated that the acetylation activity of MSL is largely restricted to H4K16 whereas NSL has a broader specificity[16]. Notably, we found that knockdown of *Msl1*, a key component of the MSL complex, resulted in reduced H4K16ac levels and downregulation of MOF-regulated gene in mouse keratinocytes. These data suggest that MSL plays an important role in mediating MOF functions in the skin. Future studies will be needed to determine whether the loss of the MSL complex, through *Msl1* cKO, recapitulates the defects of MOF cKO skin or the loss of both MSL and NSL is required for MOF functions in the skin. Collectively, our results suggest that MOF-mediated H4K16ac is essential for mammalian skin development and demonstrate epithelial cells of the skin as a model system to examine the physiological function of H4K16ac in vivo.

In search for the complete set of MOF targeted genes in the skin, we performed both scRNAseq and bulk RNAseq. Surprisingly, we have identified numerous ciliary genes, which largely escaped the detection in scRNAseq, and mitochondrial genes, which were detected in both single-cell and bulk RNAseq, as the major classes of MOF targets. Interestingly, ciliary genes are regulated by the conserved RFX TF family[44–46], and the promoter regions of MOF-regulated ciliary genes are highly enriched for the motifs of RFX2 and RFX3. This serendipitous finding led us to determine that MOF-regulated promoters, many of which are co-bound by RFX2, have significantly more accessible chromatin than those of epithelial lineage genes, supporting the notion that H4K16ac regulates decompaction of the chromatin. Interestingly, RFX2 preferentially binds to promoters (79%), in sharp contrast to ΔNp63, which prefers distal enhancers (43%) over promoters (16%). Thus, these results nominate RFX2 as a new member of the class of TFs with a strong preference for gene promoters[48]. Elucidation of the coordination between RFX2 and MOF to regulate ciliary and mitochondrial genes will provide an example of the interplay between TFs and histone H4 acetylation to maintain important cellular functions (Fig. 8f). Notably, a recent study of the NuA4 acetyltransferase complex in yeast provides a structural basis for the dual function of H4 acetylation in regulating chromatin packaging and transcription activation[49]. We speculate that MOF-mediated H4K16ac and its interaction with TFs, such as RFX2, is an intrinsic mechanism to link chromatin decompaction and transcription activation at gene promoters.

We have identified mitochondria and cilia as two important organelles that are regulated by MOF. The function of primary cilia during skin development has been extensively studied recently. Notably, their function in epidermal differentiation[40] and downgrowth of hair follicles[50,51], through the control of Shh signaling, was reminiscent to defects observed in MOF cKO skin. In addition to the widespread downregulation of ciliary genes, including genes in Shh signaling, in MOF cKO epithelial cells, we confirmed the strong reduction of primary cilia in both epidermis and hair germs. In contrast, the role of mitochondria, in particular the specific function of the ETC, during embryonic skin development is unknown. Furthermore, whether MOF regulates mitochondrial function through direct intramitochondrial mechanisms or nuclear mechanisms is still under active debate[52]. Interestingly, genetic ablation of Complex III of the ETC, which disrupts the entire ETC[33], has recapitulated the severe defects in epidermal differentiation and hair growth. Notably, previous studies have implicated MOF in controlling mitochondrial functions by activating fatty acid oxidation pathway[3] or regulating mitochondrial transcription and respiration through a subset of the NSL complex within the mitochondria[1], respectively. Our results have identified the ETC as another major mitochondrial function regulated by this

versatile layer of gene regulation. Of note, genetic deletion of *Tfam* (transcription factor A, mitochondrial), a DNA-binding protein essential for transcriptional activation of mtDNA[53], caused defects in epidermal differentiation and HF development but the animals survived more than 2 weeks after birth[54], which is less severe than MOF and QPC cKO. Therefore, MOF-mediated nuclear mitochondrial gene expression plays a major role in the skin. Future investigation will be needed to elucidate how the ETC functions, including the production of ATP, ROS, or other metabolites, promote epidermal differentiation and hair growth in a MOF-dependent manner.

In conclusion, this study provides genetic, genomic, and molecular evidence for MOF-mediated regulation of mitochondria and primary cilia and demonstrates coordination between RFX2 and MOF to govern gene expression in mammals. This research now paves the way to probe more deeply into the intricate network of mitochondria, cilia and MOF-controlled epigenetic regulation in development, homeostasis, and disease.

## Methods
### Mice
All experiments were carried out following IACUC-approved protocols and guidelines at CU Boulder and Northwestern, respectively. Mice were bred and housed according to guidelines of the IACUC in a pathogen-free facility at the University of Colorado at Boulder and at Northwestern University Feinberg School of Medicine. Housing rooms have 12-hr light/dark cycles with an ambient temperature of 23 °C and 50% humidity.

MOF fl/fl line was generated as described previously[5]. Kansl1 fl/fl line was generated from a conditional ready, targeted ESC strain obtained from the European Mouse Mutant Archive (EMMA). In this conditional allele, exon 3 (out of total 14 exons) is targeted for Cre-mediated recombination and deletion. Uqcrq (QPC) fl/fl line was described previously[33]. Other mouse lines used include: Krt14-Cre (E. Fuchs, Rockefeller University), and Krt14-H2BGFP (E. Fuchs, Rockefeller University). Seven weeks old heterozygous animals with Krt14-Cre were used for breeding to generate conditional knockout. For MOF study, pregnant females were taken at E15.5 or E16.5 for embryos; for QPC study, pregnant females were taken at E17.5 or E18.5 for embryos; for the Kansl1 study, newborn animals were examined for a phenotype. No sex related phenotype was found in this study, so both male and female animals were used for analysis without discrimination. All the mice were C57Bl/6J and CD-1 mixed background.

### Immunofluorescence and H&E staining
OCT-embedded embryos or dorsal skin tissues were sectioned to 10 µm, fixed in 4% PFA for 10 min at room temperature (RT), permeabilized with 0.1% Triton X-100 for 10 min at RT. For BrdU staining, samples were treated with 24 U/mL DNase in DNase Buffer (30 mM Tris-Cl, pH8.1, 0.5 mM β-Mercaptoethanol, 30 mM $MgCl_2$) for 1 h at 37 °C before proceeding to the next step. When staining with mouse derived antibodies, mouse-on-mouse basic kit (BMK-2202, Vector Laboratories) was used. Otherwise, blocking was performed with 5% normal serum of the same species that the secondary antibody was raised in. Sections were incubated with primary antibodies overnight at 4 °C. Next day, sections were washed three times for 5 min each, then incubated with secondary antibodies for 2 h at RT. Nuclei were stained with Hoechst33342 (1: 5,000, Invitrogen). Slides were then mounted with VECTASHIELD antifade mounting medium (Vector Laboratories, H-1000). Imaging was performed either on a Leica DM5500B microscope with an attached Hamamatsu C10600-10B camera and MetaMorph software, or a Nikon A1 confocal microscope. Primary antibodies and dilution used in this study include H4K16ac (Sigma-Aldrich, #07-329, 1:2000), Krt5 (Covance, #SIG-3475, 1:2000), Krt1 (Covance, #PRB-165P, 1:2000), Loricrin (The Rockefeller University, Gift from E. Fuchs, generated in the lab, 1:1000), β4 integrin

(BD Biosciences, #553745, 1:200), Col17 (Abcam, #ab184996, 1:500), Pericentrin (Covance, #PRB-432C, 1:200), E-Cadherin (The Rockefeller University, Gift from E. Fuchs, generated in the lab, 1:200), α-Catenin (Cell Signaling Technology, #3236, 1:1000), Active-Caspase3 (R&D Systems, #AF835, 1:1000), Sox9 (Millipore, #AB5535, 1:500), Ki67 (Abcam, #ab15580, 1:500), Lef1 (Cell Signaling Technology, #2230, 1:500), P63 (Cell Signaling Technology, #4892, 1:200), BrdU (Abcam, #ab6326, 1:500), Krt6 (Covance, #PRB-169P, 1:500), H3K4me3 (Cell Signaling Technology, #9751, 1:200), H3K27ac (Cell Signaling Technology, #8173, 1:200). Secondary antibodies used in this study include goat anti-chicken, Alexa Fluor 488 (Invitrogen, A-11039, 1:2000), goat anti-rabbit, Alexa Fluor 555 (Invitrogen, A-21428, 1:2000), goat anti-rat Alexa Fluor 555 (Invitrogen, A-21434, 1:2000), goat anti-rat Alexa Fluor 488 (Invitrogen, A-11006, 1:2000).

Wholemount staining of Arl13b (NeuroMab clone N295B/66, #AB_11000053, 1:10) in Krt14-H2BGFP+ samples was used for cilia detection. The protocol was adapted from a previous report[55]. Briefly, freshly dissected Krt14-H2BGFP+ E16.5 control and MOF cKO dorsal skin samples were fixed in 4% PFA, blocked in PB Buffer (0.5% skim milk powder, 0.25% fish skin gelatin, 0.5% Triton X-100 in 1x PBS) for 1 h at RT. One drop of M.O.M blocking reagent from the mouse-on-mouse basic kit (BMK-2202, Vector Laboratories) was added per 1 mL PB buffer during the blocking since Arl13b is a mouse antibody. Arl13b was diluted in PB buffer and the samples were incubated with primary antibody overnight. The next day, the samples were washed twice at 2 ~ 3 h each time, and then incubated with 1:1000 goat anti-mouse IgG2a Alexa Fluor 555 secondary antibody (Invitrogen, A-21137) and 1:5000 Hoechst33342 overnight. On the third day, samples were changed to PBS, without thorough washes, and mounted onto slides with VECTASHIELD antifade mounting medium. Imaging was performed on a Nikon A1 confocal microscope with the software Nikon NIS Elements, version 4.51.00.

For H&E staining, sections were fixed in 4% PFA for 10 min at RT, then dipped in hematoxylin for 3 min, and eosin for 6 sec. After going through a series of dehydration steps, sections were mounted with Permount mounting medium (Fisher Scientific, SP15-500). Imaging was performed on a Leica DM5500B microscope with a Leica camera and software Leica LAS X suite, version 3.7.

### Fluorescence image quantification

The level of cellular fluorescence from fluorescence microscopy images was determined following this protocol (https://theolb. readthedocs.io/en/latest/imaging/measuring-cell-fluorescence-using-imagej.html) using Fiji[56]. Briefly, the corrected total cell fluorescence (CTCF) is calculated using the formula: CTCF = Integrated Density − ( Area of selected cell X Mean fluorescence of background readings). Three readings from the background were averaged as the mean fluorescence background reading. The CTCF values were divided by 10,000 for plotting purposes to simplify the labeling.

For Pericentrin staining, quantification is based on the orientation relative to the basement membrane.

To quantify apoptotic (active Caspase 3 staining) or proliferative (BrdU staining) cells, positively labeled cells were counted for every image taken under 20x objective using the Leica DM5500B microscope. Every 20x image is considered one field and quantification is labeled as the number of positive cells per field.

### In situ hybridization and quantification

RNA in situ hybridization was performed using RNAscope Multiplex Fluorescent Detection Kit (Advanced Cell Diagnostics, ACD, #323110) with commercially available probes for mouse Scp2 (ACD, #875961) and Opa1 (ACD, #513961-C3) following the manufacturer's protocol. Scp2 and Opa1 were detected simultaneously on the same slide with different fluorescent reagents. Quantification of in situ was performed using the "Analyze particles" function of Fiji. And the number of

particles in each field of view is divided by the number of nuclei to get the average number of particles per nucleus.

### Flow cytometry

For E16.5 sorting, total dorsal skin was mined into small pieces and incubated with 0.1% collagenase (Worthington, LS004188) for 30 min at 37 °C. After incubation, PBS was used to dilute the collagenase, then pipet to dissociate. Tissues were pelleted by centrifuge and then subjected to fresh Trypsin digestion for 5 min at 37 °C. PBS supplemented with 5% chelated FBS was used to neutralize Trypsin and cells were filtered through a 40-μm cell strainer. DAPI-positive dead cells were excluded and epithelial cells were enriched by selecting Krt14-H2BGFP+ cells. Flow cytometry was performed on the MoFlo XDP machine (Beckman Coulter).

For P0.5 newborn epidermal cell isolation, dorsal skin was first incubated with 2.5 U/mL Dispase to separate the epidermis from the underlying dermis. Then epidermis was floated on Trypsin, incubating for 5 min at 37 °C. Single cell suspension was generated by pipetting and filtering through a 40-μm cell strainer and used for downstream experiments.

### Electron microscopy

E16.5 control and MOF cKO dorsal skin samples were fixed in EM fixation solution (2% Glutaraldehyde, 4% PFA, 50 mM Na Cacodylate, 2 mM Calcium Chloride) in a glass vial, and then bring to the CU Boulder EM Service Core Facility for further processing. Imaging was performed using the FEI Tecnai F30, 300 kV FEG-TEM system.

### shRNA infection

pGIPZ vector clone expressing control or Msl1 shRNA were obtained from Northwestern SBDRC Gene Editing, Transduction, and Nanotechnology (GET iN) core. Lentivirus were packed in HEK293T cells (ATCC, CRL-11268). Lentivirus supernatant was added to wildtype mouse keratinocytes in a 6-well plate overnight to infect. 24 h after infection, 1 μg/mL puromycin was added to the cells to select positively infected cells. 72 h after selection, cells were collected for qPCR or plated onto the cover glass for immunofluorescence staining.

### Reverse transcription and real-time PCR

Reverse transcriptions were performed using SuperScript III First-Strand Synthesis SuperMix (Invitrogen, 18080-400). Real-time PCR reagents were from Bio-Rad. Reactions were performed according to the manufacturer's manual on a CFX384 real-time system (Bio-Rad). Differences between samples and controls were calculated using the $2^{-\Delta\Delta C(t)}$ method. Real-time PCR primers were listed in Supplementary Data 5.

### RNAseq library construction and analysis

Total RNA was isolated from flow cytometry enriched E16.5 control and MOF cKO epithelial cells using TRIzol (Invitrogen), and RNA quality was assessed with Agilent 2100 bioanalyzer. 1 μg total RNA was used for RNAseq library construction using NEBNext Ultra Directional RNA Library Prep Kit for Illumina (NEB, E7420L) per the manufacturer's protocol.

For Kansl1 RNAseq, which was performed at P0.5, the epidermis was separated from the dermis after Dispase treatment, and then directly floated on 1 mL TRIzol reagent, incubated at RT for 15 min. The lysate was collected for RNA purification and library construction as described above.

RNAseq reads (150 nt, paired-end) were aligned to the mouse genome (mm10) using HISAT2 (version 2.1.0)[57]. Expression of each gene was calculated from the alignment Bam file by HTSeq-count[58]. Differentially expressed genes were determined using R package DESeq2[59] with padj <0.01, 1.5x fold-change as cutoff. Gene ontology analysis for changed genes was performed using DAVID Bioinformatics Resources

6.8[60]. GSEA[38] analysis was performed using fold change values (MOF cKO *vs.* control) as the expression dataset, and gene sets were downloaded from GSEA website for ciliary gene set and mitochondrial gene set or curated for p63 targets from a previous publication[26].

## ChIPseq and Cut&Run

All ChIPseq and Cut&Run experiments were performed using newborn total epidermal cells. For MOF ChIPseq, epidermal single cell suspension was fixed with 1% PFA for 10 min at RT, quenched with 2.5 M glycine, and followed by sonication. Sheared chromatin was pulled down using MOF antibody (Bethyl, #A300-992A, 1:100), incubated at 4 °C overnight. The next day, magnetic protein A beads (TheromFisher, #10001D) were added and further incubated at 4 °C for 2 h. After several rounds of washes, MOF and its bound DNA were eluted from the beads and subjected to reverse-crosslinking, proteinase K digestion, and DNA clean-up using QIAquick PCR purification kit (Qiagen, #24108). NEBNext® Ultra™ II DNA Library Perp Kit for Illumina (NEB, #E7645) was used for library construction. Cut&Run was conducted following this protocol: https://protocols.io/view/cut-amp-run-targeted-in-situ-genome-wide-profiling-zcpf2vn.html. Cut&Run antibodies used in this study include H4K16ac (Sigma-Aldich, #07-329), H3K4me3 (Cell Signaling Technology, #9751), H3K4me1 (Cell Signaling Technology, #9723), H3K27ac (Cell Signaling Technology, #8173), Rfx2 (Sigma-Aldrich, #HPA048969). All were used at 1:100. Briefly, epidermal cells were reversibly permeabilized with 0.05% Digitonin, bound to ConA-coated magnetic beads, 10 min at RT, and then incubated with antibody at 4 °C overnight. The next day, after several washes, Protein A-MNase fusion protein was added to the cells and incubate at 4 °C for 1 hr. After washes, $CaCl_2$ was added to activate the MNase, incubating at 0 °C for 30 min. After inactivating MNase with EGTA/EDTA, the reaction was incubated at 37 °C for 30 min to release the Cut&Run fragments from the insoluble nuclear chromatin. Proteins were removed by proteinase K digestion and DNA was purified by Phenol/Chloroform extraction. The libraries were constructed using NEBNext® Ultra™ II DNA Library Prep Kit for Illumina®.

Single-end ChIPseq and Paired-end Cut&Run reads were aligned to the mouse genome (mm10) using Bowtie2[61] with an option "−very-sensitive-local". Peak calling was performed on each individual sample by MACS2[62]. The parameter "-BAMPE" was used for Rfx2 peak calling.

*K*-means clustering was performed using seqMINER (version 1.3.4)[63]. First, mapped Bam files were converted to Bed files using bedtools, then 10 million (MOF ChIPseq and H4K16ac, H3k4me1, H3K27ac Cut&Run) or 2 million (H3K4me3) mapped reads were randomly sampled from each library and used as input. MOF ChIPseq peaks were used as genome coordinates. Signals were calculated in a 2 kb region (±1000 bp) surrounding the center of the peak with 25 nt bin.

Enriched motif search was performed using Homer[64] *findMotifGenome.pl*. Peak distribution in the genome is annotated using Homer *annotatePeaks.pl* function.

## single-cell RNAseq library generation

Single cell suspensions from two pairs of E15.5 and E16.5 control and MOF cKO total dorsal skin were used as input for single cell RNAseq. The libraries were constructed following 10x Genomics Chromium Single Cell 3′ Reagent Kits v2 or v3 User Guide (PN-120237). The libraries were sequenced on the Illumina NovaSeq 6000 platform to achieve an average of approximately 50,000 reads per cell.

For QPC scRNAseq, E17.5 control, and QPC cKO total dorsal skin were used as input. The libraries were constructed following 10x Genomics Chromium Single Cell 3′ Reagent Kits v3 User Guide (PN-1000075). The libraries were sequenced on the Illumina NovaSeq 6000 platform to achieve an average of approximately 50,000 reads per cell.

## single-cell RNAseq analysis

The 10x Genomics Cellranger Single-Cell Software Suite (version 6.1.2) was used to perform barcode processing and single-cell 3′ gene counting. Barcodes, features, and matrix files were loaded into Seurat 4.0[65] for downstream analysis. Low quality and dead cells were excluded from analysis using nFeature_RNA (>1000 and <6,000) and mitochondrial percentage (<10%). In addition, cell cycle regression was used to regress out addition variation from cell cycle genes. After UMAP dimension reduction and clustering, cluster markers were used to identify distinct cell populations. Genes with log2FC of more than 0.5 were used for GO term analysis. To analyze epithelial cell in higher resolution, we subset epithelial cells and then re-ran the analysis pipeline.

To calculate nuclear mitochondrial score, a list of 1158 mitochondrial genes downloaded from GSEA was used to calculate a module score using *the AddModuleScore* function of Seurat.

To make the cell number comparable, 600 epithelial cells were randomly subset from QPC control and cKO samples and then integrated with MOF epithelial cells using Seurat 4.0 for further analysis.

## single-cell ATACseq library generation

For scATACseq, E16.5 control and MOF cKO total dorsal skins were subject to flow cytometry to sort for Krt14-H2BGFP⁺ epithelial cells and Krt14-H2BGFP⁻ non-epithelial cells, then the GFP⁺ cells were mixed with GFP⁻ cells at 2:1 ratio to enrich for epithelial cells. The libraries were prepared using the 10x Genomics Chromium Single Cell ATAC Library & Gel Bead kit (PN-1000110). The libraries were sequenced on the Illumina NovaSeq 6000 platform to achieve an average of approximately 50,000 paired-end reads per nucleus.

## single-cell ATACseq analysis

Raw sequencing reads were aligned using cellranger-atac (version 2.0.0) with the corresponding reference genome (refdata-cellranger-arc-mm10-2020-A-2.0.0) downloaded from 10x Genomics website. The generated *filtered_peak_bc_matrix* and *fragment* files were used for further analysis.

R package ArchR[47] was used for cell clustering, TSS enrichment score calculation, and Tn5 insertion profile plotting. MOF targets were identified as genes that have MOF binding on their promoter and down-regulated in MOF cKO. Epithelial markers were identified from scRNAseq.

## Statistics and study design

RNAseq (E16.5) and scRNAseq (E15.5 and E16.5) for control and MOF cKO were performed with two pairs of samples. RNAseq (P0.5) for control and Kansl1 cKO was performed with two pairs of samples. ScRNAseq (E17.5) for control and QPC cKO was performed with one pair of samples. ScATAC-seq was performed with one pair of E16.5 control and MOF cKO samples. All ChIPseq and Cut&Run were performed in P0.5 wildtype total epidermal cells. MOF ChIPseq, H3K4me1 and Rfx2 Cut&Run have one replicate, H4K16ac Cut&Run has three replicates, H3K4me3 and H3K27ac Cut&Run have two replicates. All the experiments were designed such that there were always littermate controls. All statistical tests were performed using unpaired two-sided Student's *t*-test. The exact *P* values are shown in Supplementary Data 4 For boxplot, the lower and upper hinges correspond to the first and third quartiles, and the middle line represents the median. The upper whisker extends from the hinge to the largest value no further than 1.5 * IQR from the hinge (where IQR is the inter-quartile range, or distance between the first and third quartiles). The lower whisker extends from the hinge to the smallest value at most 1.5 * IQR of the hinge. Data beyond the end of the whiskers are plotted individually.

## Reporting summary

Further information on research design is available in the Nature Portfolio Reporting Summary linked to this article.

## Data availability

All sequencing data have been deposited in NCBI's Gene Expression Omnibus and are accessible through GEO Series accession number GSE214441. Mouse genome mm10 is available at http://genome.ucsc. edu/cgi-bin/hgGateway?db=mm10. Mouse cilium gene set was downloaded at https://www.gsea-msigdb.org/gsea/msigdb/mouse/geneset/ GOCC_CILIUM.html. Mouse mitochondrion gene set was downloaded at https://www.gsea-msigdb.org/gsea/msigdb/mouse/geneset/GOCC_ MITOCHONDRION.html. Mouse reference dataset for scRNAseq was downloaded from https://support.10xgenomics.com/single-cell-gene-expression/software/downloads/latest. Mouse reference dataset for scATACseq was downloaded from https://support.10xgenomics.com/ single-cell-atac/software/downloads/latest. Source data are provided with this paper.

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

## Acknowledgements

We thank E. Fuchs (Rockefeller University, HHMI) for Krt14-H2BGFP and Krt14-Cre mice, J. Wallingford (University of Texas at Austin) for suggestions regarding to ciliary studies, P. Bhalla and I. Budunova (Northwestern University SBDRC, P30AR075049) for constructs, and all members of the Yi laboratory for suggestions. Confocal imaging work was performed at the Northwestern University Center for Advanced Microscopy generously supported by NCI CCSG P30 CA060663 awarded to the Robert H Lurie Comprehensive Cancer Center. This work was supported by National Institute of Health Grant R01AR071435, R01AR081103 and R01HD107841 (RY).

## Author contributions

D.W. performed most experiments and analyses with assistance from H.L.; H.L. performed ChIP-seq and Cut&Run profiling; N.S.C. provided the QPC mouse model and advised the mitochondrial study; Y.D. provided the MOF mouse model and advised the MOF/H4K16ac study; R.Y. conceived and supervised the study and wrote the manuscript together with D.W. All authors participated in the manuscript preparation.

## Competing interests

The authors declare no competing interests.
