## [Peer Review File · Nature Communications]

MOF-mediated Histone H4 Lysine 16 Acetylation Governs Mitochondrial and Ciliary Functions by Controlling Gene PromotersREVIEWER COMMENTS

Reviewer #1 (Remarks to the Author):

Wang et al., attempt to establish the role of MOF-dependent H4K16ac in epidermal development. The role of this important histone mark in this essential tissue has never been addressed to my knowledge and it is as such interesting to both the epigenetics and the developmental biology field. They used Krt14-Cre to drive epidermis-specific MOF (the enzyme essential for this mark) knockout and demonstrate that epidermal differentiation, cell adhesion and hair follicle development were severely affected by H4K16ac depletion. The phenotype is strong and exciting and convincingly documented in this paper, which is a strength of the manuscript. With genomic tools such as scRNA-seq, bulk RNA-seq, scATAC-seq and CUT&RUN, they provide evidence that mitochondria and primary cilia dysregulation may contribute to the observed MOF cKO phenotype. The mechanistic data and analysis pose some technical caveats and need more strengthening in this reviewer's opinion. Overall, there is a lot of genomics data, but it appears somewhat superficial in both rigor of its collection and of its analysis. The story around it is a little choppy and could be a bit better developed. Perhaps fewer datasets of higher quality with more in-depth analysis and some in situ validation may improve the manuscript.

Specific comments are provided below.

The RNA-seq performed on total epithelial cells at E16.5 is somewhat problematic because there is already a strong phenotype with fewer suprabasal cells and missing hair follicle by that time point. So, a lot of the changes are likely due to different cell-type composition of the sample, and not due to actual changes in gene expression. One solution would be sorting basal cells away from hair follicle and from suprabasal cells based on cell surface markers, another would be using an earlier time point prior to phenotype onset to do RNA-seq.

The scRNA-seq would have been a good approach in clarifying the target genes despite the phenotype onset by E16.5. Unfortunately, that scRNA-seq data has its own caveats. There are very few cells probed in total by scRNAseq : 5656 in 4 samples 2 KO and 2 WT, and with 2-5% epithelial cells in cKO we are probably looking at a minuscule number of epithelial cells that the authors even subcluster further in basal and suprabasal; they had no hair placode cells in the KO even though we know they are present. This is concerning because low cell number may skew the analyses and important genes could be missed. In fact, the cilium genes, which ended up being the most interesting ones from the RNA-seq data were not detected in scRNA-seq of basal cells, maybe because they are strongly expressed only in hair germ cells...? I would suggest performing scRNA-seq with FACS-sorted epithelial cells -even better at E15.5 prior to phenotype onset - to catch the early changed genes and a good number of cells and avoid problems with hair follicle and suprabasal cell number phenotypes. It would be a big difference with 10-20k starting cells.

With this concern above said, the presence of MOF on promoters in ChIP-seq experiments on putative target genes identified above in the RNA-seq data provides some comfort that those target genes are probably true targets, but it'd be good to strengthen the transcriptomics data one way or another and maybe validate some important gene in situ.

There seems to be a defect in cilium formation as shown in Figure 4, so at least the RNA-seq sequencing data might have been useful to pick up important changes. But is that due to the cilium itself lacking or is due to the lack of hair germs, where the authors say cilium is strongly represented or due to delayed epidermis development? The authors say placodes do form OK, so maybe showing how quantification of cilium was done in the hair placodes vs in the basal layer of the epidermis – and demonstrating that it is indeed missing from specific skin structures and we are not looking at a problem of lacking hair germs in fact. This would separate the hair follicle phenotype from the cilium phenotype and make it more convincing. Again in this study, none of the target genes have been validated in vivo, an approach that could strengthen the validity of the mechanism.

The authors draw a conclusion that MOF affects Shh signaling (Figure 7h), but there is no direct evidence of it aside from the somewhat questionable cilium data above (see above point). Is there a way to show shh pathway activation was affected directly in situ? If not, perhaps shh target genes are down in the transcriptomics data?

The Krt5high/Krt1low subcluster from cKO scRNA-seq data is very interesting. In UMAP space, this subcluster is far from basal cells and closer to the other differentiated cluster, so the Krt5+ cells in Figure 2f might actually be differentiated cell instead of delaminated basal cells. It would be nice to go deeper into this cluster, looking at what genes are expressed (more basal or more suprabasal) and how they are different from the other suprabasal cluster in terms of gene expression. With better-quality scRNA-seq data, this would be very meaningful.

The scATAC-seq data should be mined in more depth. I do see clusters in Extended data 5b but not much text around that. The identity of those clusters is also not defined. Basal and suprabasal cells should already be different at chromatin level and I am wondering whether this is still true for cKO samples. There are tools available for combined scRNA-seq and scATAC-seq analysis which might be helpful to reveal the identify of those scATAC-seq clusters and provide more insight into how cell differentiation is regulated by MOF-dependent chromatin accessibility.

Regarding ETC and MOF. There are probably many mutants that show similarities in phenotypes and molecular changes, but I don't think that is immediately genetic and molecular evidence of their interaction, as the authors imply here. Maybe I'm missing something in Figure 7 that can be explained better, but I find these data distracting and not that useful as I was not convinced that the few genes controlled by ETC that change in MOF, is enough evidence to connect these two mechanistically.

Minor concerns

The high-quality cell number eventually recovered from scRNA-seq and scATAC-seq for each sample should be explicitly specified somewhere in the manuscript, either in main context, figure legend, or methods, especially the number of cells used for cluster analysis.

Please spell out MOF cKO on figures instead of just cKO, which is confusing, since several cKO exist in this paper

Reviewer #2 (Remarks to the Author):

MOF-mediated Histone H4 Lysine 16 Acetylation Governs Mitochondrial and Ciliary Functions By Controlling Gene Promoters, Wang et al.

The authors study the role of MOF in skin development using conditional deletion of Mof in the epidermis. They use a variety of histological and genomics techniques to assess the role of MOF. MOF is the enzymatic component of two distinct complexes the MSL complex, which has the primary function of acetylating H4K16 and the NSL complex which has the primary function of acetylating other H4 lysines, predominantly K5 and K8.

The authors make a number of claims which are poorly supported by the data presented. The experiments in general are well performed and involve technically difficult techniques for example, single cell RNA-seq, ATAC-seq etc. However, the overall experimental design is lacking and results, I believe, are over-interpreted.

1. The authors, at least in the abstract, pose the question of, what are the different functions of the two MOF containing complexes. In essence they compare the phenotype of the MOF cKO with the KANSL1 cKO. MOF has a severe phenotype in the skin; KANSL1 a minor or no phenotype in the skin.

However, since MOF is found in both complexes no conclusion can be drawn as to what the relative importance of MSL vis a vis NSL is in the skin. This is a worthwhile question and to answer it the authors should KO an essential component of the MSL complex and compare the phenotype to a KO of an essential component, presumably KANSL1, of the NSL complex. Additionally, they should first establish that the NSL complex, i.e. all the components, of the NSL complex are expressed in the epidermis and then show in the NSL-specific KO that the relevant histone acetylation marks are reduced.

2. The major part of the paper is the analysis of MOF cKO in the epidermis using the Cre- recombinase driven by Keratin 14 promoter. The analysis is then done using cells isolated at E16.5. The major problem with these experiments is that this is examining the end stage of the effects of loss of MOF function. By this point the cells are no longer proliferating (no BrdU labelling) and are either dead, dying or senescent. The epidermis has high levels of activated Caspase 3, around 25-30 cells per field (Extended data fig 2. How many cells in a field? Is the field shown in (c)? Is this E16.5 as implied by the results section?). Activated Caspase3 only marks a small proportion of apoptotic cells as this is a short-lived part of the cascade. So, it is likely that a very high proportion of the cells have undergone apoptosis as soon as the H4K16ac mark dropped below a critical threshold. The cell nuclei although difficult to see at this magnification appear to have clumped chromatin. This suggests the phenotype in these cells resemble the phenotype reported for the germline deletion (Thomas MCB 2008). Overall, the remaining part of the paper is using genomics techniques that by and large are being used to study the end point of apoptosis. If this is just a cell-autonomous lethal phenotype, then it is not surprising that it resembles other cell-autonomous lethal phenotype, for example loss of electron transport chain complexes.

3. The authors use ssRNA-seq to show that in the MOF KO dermis inflammatory gene expression is up, which is not surprising given the extent of apoptosis in this tissue and mitochondrial gene expression is down. If the cells are not proliferating, and probably dying, it is hardly surprising that their energy requirements are reduced. Likewise, it is not surprising that other cellular functions are also compromised as a result, including ciliary gene expression. I note that p21, a p53 target gene regulating cell cycle arrest is one of the significantly upregulated genes. The authors have not shown that these are specific MOF regulated genes rather than just a non-specific consequence of major disruption to chromatin organisation.

4. The authors then examine the location of MOF and various histone marks in the genome, including H4K16ac. As the authors state this really just confirms published data. This is, however, something of a missed opportunity. While H4K16ac is enriched at TSS it must also be widely distributed throughout the genome. Studies have shown that somewhere between 10-30% of nucleosomes are decorated with H4K16ac (Hansen NatComm 2019), so clearly the peaks shown in this work are just enrichment. Since the authors have MOF KO cells (ie. H4K16ac deficient) they could have used these to determine the real level of H4K16ac and MOF across the genome rather than use peak calling, which more or less assumes most intergenic regions are negative. The authors claim there is a specific function of MOF in regulating Rfx2 target genes. They find 4366 MOF binding peaks at promoters, which overlap with 47% of Rfx2 bound genes. Is this statistically significant? 4366 promoters are a substantial percentage of the expressed genes in a cell and given that the peak calling has a rather arbitrary cut-off, since there is no negative control in this experiment, so it is entirely possible that any group of 300 randomly selected expressed genes would show the same enrichment for MOF at their promoters. Also, the number of repeats of these experiments is not stated in the results or methods section.

Reviewer #3 (Remarks to the Author):

In this paper, "MOF-mediated Histone H4 Lysine 16 Acetylation Governs Mitochondrial and Ciliary

Function By Controlling Gene Promoters," Wang et al. have identified the histone acetyltransferase, MOF, and its corresponding marker H4K16ac control the mitochondria function and cilia formation to regulate the skin epidermis and hair follicle development. It is hypothesized that MOF and H4K16ac play a fundamental role in tissue development by regulating cellular functions and lineage differentiation during tissue development. In brief, the authors used the conditional depletion of MOF in epidermal basal cells at the embryonic stage. Loss of the H4K16ac specific in the basal cells upon the MOF cKO, they found that several epidermis defect phenotypes and barrier function leads to perinatal death. The authors used the scRNAseq, or bulk RNAseq, combined with the MOF and histone mark Cut and Run results, they identified the mitochondria and primary cilia genes are regulated by MOF-H4K16ac. Moreover, they used genetic deletion of QPC, one of the subunits of the mitochondria electron transport chain (ETC) complex, to phenocopy the epidermis development defect similar to MOF cKO. Overall, the authors reached three major conclusions: 1) NSL complex does not participate in the MOF-mediated regulation of tissue development 2) MOF-H4K16ac changed chromatin accessibility and regulated genes are important for mitochondria function and primary cilia formation. 3) Role of MOF-regulation within mitochondria is vital for epidermal differentiation and hair follicle formation. The results clearly support the conclusions; however, minor additions may need to be considered for a more convincing argument.

Major points:

1. Is it being suggested that MSL is the complex which mediates MOF function during mammalian tissue development? The argument against NSL is clear, as the authors found no major differences in mammalian development with a cKO of *Kansl1*, an important component of NSL. However, there is no evidence in support of the MSL complex, so the initial question of "whether the function of MOF is mediated by MSL or NSL" is still open. Does MSL mutant mouse also have similar phenotype? My suggestion would be to show that MSL is important for MOF function during development. In figure 1c, the IF image of *Msl1* knockdown by shRNA in keratinocyte is also not clear.
2. Were there any functional analyses done for MOF function in primary cilia? Especially since the cKO QPC does not address some of the observed phenotypes from the MOF cKO. It seems that the function of MOF in this context was overlooked in addition to role of MOF in primary cilia. Also, it lacks evidence to show that the MOF cKO-primary cilia axis affects skin development.
3. The authors performed the bulk RNAseq by FACS purification cells. Can authors show the sorting profiles? Does the MOF cKO affect epithelial markers expression?
4. Is the H3K4me3 the global reduction or mitochondria/primary cilia genes specifically in MOF cKO?

Minor points

1. The authors should include detailed information about the quantification method in the material and method. It is also unclear in the figure; What is the unit in quantifying immunostaining?
2. It would be beneficial for the conclusion that there is a reduction in HF formation in MOF cKO mice (Figure 2a), to have quantitative data like what was done in Figure 7b.
3. What is the age of *Knas1* cKO in figure 1d-f?
4. In line 145, the authors mention "E18.5 epidermal detachment became widespread, and the reduction of basal cells..." How do the authors get the conclusion about the reduction of basal cells from figure 2a? Are there any markers staining and quantification results to indicate that reduce the basal cells in MOF cKO epidermis at E18.5?
5. In figure 2e, the E-cad and α -Cat cannot clearly see the distribution of these two markers change in control and MOF KO.
6. In line 248, "MOF and H4K16ac have been implicated in the regulation of "housekeeping" genes..." Please cite the references in this sentence.
7. In figure 4e, what is the "HG" in the ctrl skin? The primary cilia are enriched in the HG in ctrl skin. Is there any HG in the MOF cKO skin?
8. In figure 6, Can authors show the profiles of the MOF target genes such as mitochondria and primary cilia genes in control and MOF cKO? It would be much clearer to see the chromatin accessibility changes between control and cKO skin.

List of new/revised figures:

1. Figure 1c: new images from *Msl1* knockdown keratinocytes to show the downregulation of H4K16ac.
2. Figure 2a-2b: 2a, new H&E image of E15.5 skin, relocate the H&E result from the later time point E18.5 to Extended Data fig. 2a. 2b, new Krt5/b4 integrin image for E15.5 skin, showing the basal localization of beta4 integrin at this earlier time point, relocate the Krt5/Col17 result (E16.5) to Extended Data fig. 2c.
3. Extended Data fig. 2b: new Krt5/Krt1 image at E18.5, showing the depletion of basal progenitor cells at this later stage in MOF cKO.
4. Extended Data fig. 3a, 3c: new Ki67 results from E15.5 skin (3a), showing the universal Ki67+ signals in basal cells; new apoptosis analysis from E15.5 skin (3c), showing much less apoptotic cells in MOF cKO skin, compared with the result from E16.5.
5. Figure 3a-d: new results of E15.5 scRNAseq from 2 pairs of control and MOF cKO skin. Please see our point-to-point response and the manuscript for detailed description and analysis of these results.
6. Extended Data fig. 4a-e: new results from E15.5 scRNAseq, showing normal cell cycle progression (4c); and mosaic deletion of H4K16ac in MOF cKO6 (4e).
7. Figure 4a-d: 4a, new single-molecule in situ hybridization (RNAscope) results from two selected genes to validate scRNAseq result; 4b, new qPCR results showing the downregulation of MOF targeted genes in *Msl1* KD keratinocytes; 4c, new mitochondrial score results from E15.5 scRNAseq data; 4d, new heatmap showing the widespread downregulation of ETC genes in both basal and suprabasal cells from E15.5 scRNAseq data.
8. Extended Data fig.6b: new heatmap to show the widespread downregulation of mitochondrial genes in both basal and suprabasal cells from E15.5 scRNAseq data.
9. Figure 6e-g: 6e, new violin plots from E15.5 scRNAseq data, showing the downregulation of several detectable ciliary genes in MOF cKO; 6f, new wholemout IF results from E15.5 skin, showing robust detection of primary cilia in both basal progenitors and hair germ and the absence of primary cilia in MOF cKO; 6g, quantification of the reduction in the number of primary cilia in MOF cKO.
10. Extended Data fig.7f: new analysis of *Krt5^{hi}/Krt1^{lo}* cell cluster in MOF cKO, together with control and QPC cKO, showing the unique presence of these cells in MOF cKO.
11. Extended Data fig.8a, 8c: Flow cytometry plot for sorting H2bGFP+ epithelial cells from control and MOF cKO skin (8a); Downregulation of genes in *Shh* signaling pathway in MOF cKO (8c).

12. Figure 8d-e: 8d, two examples of MOF targeted promoters with reduced open chromatin signature in MOF cKO; 8e, one example of intact open chromatin signature in a gene promoter that is not targeted by MOF.

Reviewer #1 (Remarks to the Author):

Wang et al., attempt to establish the role of MOF-dependent H4K16ac in epidermal development. The role of this important histone mark in this essential tissue has never been addressed to my knowledge and it is as such interesting to both the epigenetics and the developmental biology field. They used Krt14-Cre to drive epidermis-specific MOF (the enzyme essential for this mark) knockout and demonstrate that epidermal differentiation, cell adhesion and hair follicle development were severely affected by H4K16ac depletion. The phenotype is strong and exciting and convincingly documented in this paper, which is a strength of the manuscript. With genomic tools such as scRNA-seq, bulk RNA-seq, scATAC-seq and CUT&RUN, they provide evidence that mitochondria and primary cilia dysregulation may contribute to the observed MOF cKO phenotype. The mechanistic data and analysis pose some technical caveats and need more strengthening in this reviewer's opinion. Overall, there is a lot of genomics data, but it appears somewhat superficial in both rigor of its collection and of its analysis. The story around it is a little choppy and could be a bit better developed. Perhaps fewer datasets of higher quality with more in-depth analysis and some in situ validation may improve the manuscript.

We thank reviewer #1 for her/his support and constructive suggestions. During the revision, we have focused on mechanistic analysis to better link compromised gene expression, related to mitochondria and primary cilia, to the observed phenotypes. Specifically, we have generated new, high-quality scRNAseq datasets from 2 pairs of control and MOF cKO mice at E15.5, when MOF and H4K16ac are significantly reduced (but not depleted) and phenotypical defects were relatively minor. This allowed us to profile many more epithelial cells and detect many transcripts. This new dataset, together with better analysis and confirmative studies, such as in situ hybridization for MOF targets, has improved our manuscript and strengthened our conclusion.

Specific comments are provided below.

The RNA-seq performed on total epithelial cells at E16.5 is somewhat problematic because there is already a strong phenotype with fewer suprabasal cells and missing hair follicle by that time point. So, a lot of the changes are likely due to different cell-type composition of the sample, and not due to actual changes in gene expression. One solution would be sorting basal cells away from hair follicle and from suprabasal cells based on cell surface markers, another would be using an earlier time point prior to phenotype onset to do RNA-seq.

We wholeheartedly agree with the reviewer that our E16.5 data should be strengthened. It was generated at the early phase of the project such that the 10x v2 kit was used, which has less robust capture of transcripts. In addition, the severe phenotype indeed caused the difficulty to 1) recover enough cells; and 2) distinguish direct vs indirect effects of MOF cKO on the transcriptome. During the revision, we initially considered the sorting experiments as suggested. However, we think that sorting may cause additional stress to already fragile cells and lead to non-specific gene expression changes. In addition, MOF cKO cells show strong defects in integrin gene expression, which may also interfere with sorting strategies if we would sort basal and suprabasal cells specifically. We concluded that scRNAseq at an earlier time point is the best option. We then generated 4 new scRNAseq libraires from 2 pairs of control and MOF cKO at E15.5. The results are very nice and provide many important insights to strengthen our study (see below for details). We thank the reviewer for this excellent suggestion.

The scRNA-seq would have been a good approach in clarifying the target genes despite the phenotype onset by E16.5. Unfortunately, that scRNA-seq data has its own caveats. There are very few cells probed in total by scRNAseq : 5656 in 4 samples 2 KO and 2 WT, and with 2-5% epithelial cells in cKO we are probably looking at a minuscule number of epithelial cells that the authors even subcluster further in basal and suprabasal; they had no hair placode cells in the KO even though we know they are present. This is concerning because low cell number may skew the analyses and important genes could be missed. In fact, the cilium genes, which ended up being the most interesting ones from the RNA-seq data were not detected in scRNA-seq of basal cells, maybe because they are strongly expressed only in hair germ cells...? I would suggest performing scRNA-seq with FACS-sorted epithelial cells -even better at E15.5 prior to phenotype onset - to catch the early changed genes and a good number of cells and avoid problems with hair follicle and suprabasal cell number phenotypes. It would be a big difference with 10-20k starting cells.

The reviewer was spot on that a better scRNAseq experiment is needed to strengthen our study. In the revision, we made 4 new scRNAseq with E15.5 samples, 2 control and 2 MOF cKO. In these new datasets, we recovered 35,361 cells (7 times more than our E16.5 dataset), among which 7,469 were epithelial cells (ctrl2: 30.88% were epithelial cells; ctrl 3: 32.05% were epithelial cells; cKO5: 9.72% were epithelial cells; cKO6: 14.78% were epithelial cells). Thus, we have profiled a large number of both control and MOF cKO cells at E15.5 when the depletion of H4K16ac is ~70% and phenotypical defects just began discernible. Interestingly, one of the cKO (cKO6) is mosaic deletion of MOF, as shown by H4K16ac staining pattern (new Extended Data fig. 4e), likely due to mosaic expression of Cre. Indeed, in the scRNAseq UMAP plot, 1,084 cKO6 epithelial cells clustered with control, while 289 epithelial cells clustered with cKO5, which was a complete deletion of MOF (new Fig. 3b and Extended Data fig. 4b). This result indicates that the effect we observed in MOF cKO is cell autonomous rather than systemic. Furthermore, the bimodal distribution of control and cKO cells and the lack of intermediate cells in between suggest the cKO transcriptome is specifically perturbed as

the result of MOF deletion but not a gradual and non-specific death signature. This is important for our target analysis.

From these new datasets, we found that: 1) the significantly downregulated genes were still the same GO categories of mitochondrion, cell cycle, proteasome, et al, (new Fig. 3c-d), confirming our previous findings from E16.5 datasets; 2) the same categories of downregulated genes were detected in basal and suprabasal cells, indicating the consistent function of MOF/H4K16ac; 3) even though HG structures in cKO5 (the complete MOF cKO) were clearly visible in H2BGFP wholemount imaging, no HG cluster could be distinguished in cKO scRNAseq. This was likely due to the significant transcriptomic changes caused by the loss of MOF and H4K16ac; 4) genes associated with primary cilia were still mostly undetectable in both control and cKO. However, with the increased capture rate by 10x v3 kit and deep sequencing of our scRNAseq libraries, we detected a few ciliary genes, including B9d1, Cfap36, Ift74 and Kif3a, all of which were significantly downregulated in MOF cKO (new Fig. 6e). These results further corroborated our bulk RNAseq results and validated one of the most novel and important findings of our study – MOF and H4K16ac control ciliary gene expression.

With this concern above said, the presence of MOF on promoters in ChIP-seq experiments on putative target genes identified above in the RNA-seq data provides some comfort that those target genes are probably true targets, but it'd be good to strengthen the transcriptomics data one way or another and maybe validate some important gene in situ.

Following the reviewer's suggestion, we have performed RNAscope experiment for two MOF targeted genes, Scp2 and Opa1, and confirmed their downregulation in MOF cKO epidermis at E15.5 (new Fig. 4a). Furthermore, we have performed qPCR analysis for these genes in Msl1 knockdown primary keratinocytes and validated their downregulation. Finally, as indicated above the new, high-quality scRNAseq data at E15.5 also validated the downregulated genes upon MOF deletion.

There seems to be a defect in cilium formation as shown in Figure 4, so at least the RNA-seq sequencing data might have been useful to pick up important changes. But is that due to the cilium itself lacking or is due to the lack of hair germs, where the authors say cilium is strongly represented or due to delayed epidermis development? The authors say placodes do form OK, so maybe showing how quantification of cilium was done in the hair placodes vs in the basal layer of the epidermis – and demonstrating that it is indeed missing from specific skin structures and we are not looking at a problem of lacking hair germs in fact. This would separate the hair follicle phenotype from the cilium phenotype and make it more convincing. Again in this study, none of the target genes have been validated in vivo, an approach that could strengthen the validity of the mechanism.

We thank the reviewer's careful comments. To strengthen our study, we use IF to detect primary cilia in E15.5 control and MOF cKO samples. As shown in new Fig. 6f, primary cilia were robustly detected in both interfollicular epidermis and HG in control and largely absent in both compartments of MOF cKO. Quantification and statistical analysis

confirmed the finding. In addition, as stated above, we can now detect a few ciliary genes in our new E15.5 scRNAseq data. Consistent with the comparable presence of primary cilia in both IFE and HG, these ciliary genes were expressed at a similar level in IFE and HG (new Fig. 6e) in control samples. However, their expression was largely depleted in MOF cKO, consistent with significantly reduced primary cilia in MOF cKO skin.

The authors draw a conclusion that MOF affects Shh signaling (Figure 7h), but there is no direct evidence of it aside from the somewhat questionable cilium data above (see above point). Is there a way to show shh pathway activation was affected directly in situ? If not, perhaps shh target genes are down in the transcriptomics data?

We thank the reviewer's comment. We apologize that we missed the opportunity to present our results more clearly. In addition to our original data obtained from E16.5 scRNAseq, we further analyzed the expression of Shh, Gli1, Ptch1 and Ptch2 in our new E15.5 scRNAseq data. The latter three are regulators or effectors as well as targets of Shh pathway. Consistent with defective Shh signaling, the expression of all 4 genes was significantly downregulated in MOF cKO (new Extended Data fig. 8c and shown here). Because Shh signaling is strongly linked to ciliary functions (reviewed by Reiter and Leroux, Nature Review Molecular Cell Biology 2017), including the localization of Ptch1 to primary cilia to regulate Shh signaling (Rohatgi et al., Science 2007), we believe that our new data strengthen the conclusion that MOF is critical for ciliary functions and Shh signaling.

The Krt5high/Krt1low subcluster from cKO scRNA-seq data is very interesting. In UMAP

space, this subcluster is far from basal cells and closer to the other differentiated cluster, so the Krt5+ cells in Figure 2f might actually be differentiated cell instead of delaminated basal cells. It would be nice to go deeper into this cluster, looking at what genes are expressed (more basal or more suprabasal) and how they are different from the other suprabasal cluster in terms of gene expression. With better-quality scRNA-seq data, this would be very meaningful.

We have analyzed this population by comparing their transcriptome to Krt1 high (canonical) suprabasal cells in control. Consistent with their Krt5 high state, cell adhesion and basal genes, including Cd44, Cdh3, Itga6, Itgb4, were elevated in these cells. Also consistent with their compromised differentiation, epidermal differentiation genes, including Krt1, Krt10, Tgm1, Tgm3, were downregulated. These gene signatures are consistent with the idea that these basal-like cells prematurely delaminated from the basal layer (hence relatively high expression of basal adhesion genes) and yet failed to embark on robust epidermal differentiation (hence relatively low expression of epidermal differentiation genes). We would like to point out that premature delamination is likely caused by the compromised basement membrane formation such that basal cells in MOF cKO are unable to anchor to the basal layer. Indeed, when we further clustered control, MOF cKO and QPC cKO together, we only detected this cluster in MOF cKO but not in control and QPC cKO (new Extended Data fig. 7f). In addition, the suprabasal cells in QPC cKO did have compromised epidermal differentiation but not compromised basement membrane (shown in Fig. 5f-g and Extended Data fig. 7b). These scRNAseq data are further supported by IF analysis, where we observed Krt5^{hi}/Krt1^{lo} suprabasal cells only in MOF cKO (Fig. 2f) but not in QPC cKO despite reduced Krt1 and Lor expression in suprabasal cells (Fig. 5c). Collectively, these data suggest that MOF cKO epidermal cells have defects in both basal cell adhesion and suprabasal differentiation whereas QPC cKO epidermal cells only have defects in suprabasal differentiation. We have clarified these points in our revised manuscript. And we thank the reviewer's suggestion.

The scATAC-seq data should be mined in more depth. I do see clusters in Extended data 5b but not much text around that. The identity of those clusters is also not defined. Basal and suprabasal cells should already be different at chromatin level and I am wondering whether this is still true for cKO samples. There are tools available for combined scRNA-seq and scATAC-seq analysis which might be helpful to reveal the identify of those scATAC-seq clusters and provide more insight into how cell differentiation is regulated by MOF-dependent chromatin accessibility.

Thank the reviewer for this suggestion. We tried to integrate scRNAseq data with scATACseq data. However, unlike our scATACseq data from adult skin, where each cell cluster can be easily distinguished (Zhang et al., Nature Aging 2021), similar to scRNAseq results, scATACseq data from embryonic skin couldn't be effectively clustered either based on open chromatin signatures or based on transcriptional activity - transcriptional activity of each gene is estimated by quantifying scATACseq fragment counts in the 2-kb upstream region and gene body. And this is the foundation for integrating scATACseq data with scRNAseq data. This result indicates a high similarity

of the open chromatin landscape among epithelial cells at this early developmental stage. Indeed, when we only used open chromatin to cluster epithelial cells in control samples, we could only distinguish basal epithelial cells and hair germ, but not the suprabasal epidermis. In MOF cKO samples, the identities of each cluster were not clear either. Thus, the integration was not successful. As a result, we used all epithelial cells from control and MOF cKO to perform the analysis of open chromatin with a focus on gene promoters. Because downregulated genes in MOF cKO basal and suprabasal cells are generally similar, this approach should still provide reliable insights into how the loss of MOF perturbs the open chromatin status of gene promoters.

Regarding ETC and MOF. There are probably many mutants that show similarities in phenotypes and molecular changes, but I don't think that is immediately genetic and molecular evidence of their interaction, as the authors imply here. Maybe I'm missing something in Figure 7 that can be explained better, but I find these data distracting and not that useful as I was not convinced that the few genes controlled by ETC that change in MOF, is enough evidence to connect these two mechanistically.

We thank the reviewer's careful comments. We agree with the reviewer that the ETC defects are not the only defects caused by MOF cKO. However, MOF cKO leads to numerous mitochondrial gene downregulation and many of which specifically function in the ETC, as further supported by our E15.5 scRNAseq data. In our unpublished studies on mitochondrial metabolism, we have knocked out several critical genes in the TCA cycle and different ETC complexes (Complex I and Complex II), in addition to the Complex III QPC knockout. None of these knockouts caused lethality at birth, unlike MOF and QPC cKO. Mechanistically, deletion of Complex III is predicted to cause the most widespread defect in mitochondrial functions – it should largely block the TCA cycle, Complex I and Complex II. Therefore, QPC cKO (loss of Complex III) most likely mimics the strong defects observed in MOF cKO.

In support of this hypothesis, both our phenotypical analysis and scRNAseq analysis revealed considerable similarity between MOF and QPC cKO (new Fig. 5). To minimize confusion, we have carefully described the defects in QPC cKO, and noted that these QPC cKO has largely intact basement membrane and normal proliferation (Extended Data fig. 7), distinct from MOF cKO phenotypes. The similarity between MOF and QPC cKO lies in epidermal differentiation as well as similar upregulation in ribosomal genes. Taken together, we believe that the molecular and genetic evidence to link MOF and the ETC function is well supported by our data.

Minor concerns

The high-quality cell number eventually recovered from scRNA-seq and scATAC-seq for each sample should be explicitly specified somewhere in the manuscript, either in main context, figure legend, or methods, especially the number of cells used for cluster analysis.

We have specified the number of cells used for scRNAseq analysis in supplement table 1 and the number of cells used for scATACseq in the main text.

Please spell out MOF cKO on figures instead of just cKO, which is confusing, since several cKO exist in this paper

We have annotated all MOF cKO, Kansl1 cKO, QPC cKO specifically in each place these models are cited throughout the manuscript.

Reviewer #2 (Remarks to the Author):

MOF-mediated Histone H4 Lysine 16 Acetylation Governs Mitochondrial and Ciliary Functions By Controlling Gene Promoters, Wang et al.

The authors study the role of MOF in skin development using conditional deletion of Mof in the epidermis. They use a variety of histological and genomics techniques to assess the role of MOF. MOF is the enzymatic component of two distinct complexes the MSL complex, which has the primary function of acetylating H4K16 and the NSL complex which has the primary function of acetylating other H4 lysines, predominantly K5 and K8.

The authors make a number of claims which are poorly supported by the data presented. The experiments in general are well performed and involve technically difficult techniques for example, single cell RNA-seq, ATAC-seq etc. However, the overall experimental design is lacking and results, I believe, are over-interpreted.

We thank the reviewer's constructive criticism. This study was the result of accumulative investigations of more than 7 years and contained a lot of data covering phenotypical analysis, transcriptomic analysis, genomic analysis and our attempt to connect molecular mechanism of MOF-regulated gene expression to the observed phenotypes. In the revision, we have focused on clarification of our results, explanation of our rationale and conclusions with experimental data. In particular, we have addressed the reviewer's main concern – whether the perturbed transcriptome is a general consequence of severe defects or is specifically caused by the loss of MOF in MOF cKO (see below for details).

1. The authors, at least in the abstract, pose the question of, what are the different functions of the two MOF containing complexes. In essence they compare the phenotype of the MOF cKO with the KANSL1 cKO. MOF has a severe phenotype in the skin; KANSL1 a minor or no phenotype in the skin. However, since MOF is found in both complexes no conclusion can be drawn as to what the relative importance of MSL vis a vis NSL is in the skin. This is a worthwhile question and to answer it the authors should KO an essential component of the MSL complex and compare the phenotype to a KO of an essential component, presumably KANSL1, of the NSL complex. Additionally, they should first establish that the NSL complex, i.e. all the components, of the NSL complex are expressed in the epidermis and then show in the NSL-specific KO that the relevant histone acetylation marks are reduced.

We thank the reviewer for raising this important question. Our first goal was to study the function and mechanism of MOF in the skin. We were also aware of the functional importance and versatility of the NSL complex, based on several prominent publications (Chatterjee et al., Cell 2016; Li et al., Molecular Cell 2009; and reviewed by Sheikh et al., EMBO Rep 2019). In contrast, the MSL complex has been suggested only partially responsible for H4K16ac, and the other part has been credited to the NSL. To examine the function of MOF and NSL, we generated both MOF and *Kansl1* cKO models with the same *Krt14-Cre*. Surprisingly, we observed very strong defects in MOF cKO whereas *Kansl1* cKO is largely intact. The effective deletion of *Kansl1* is well documented by our genotyping, qPCR and RNAseq. The lack of phenotype in *Kansl1* is also consistent with intact H4K16ac levels, determined by both Western blot and IF staining, and lack of transcriptome changes determined by RNAseq. Thus, our data suggest that *Kansl1* is not required for mouse embryonic skin development. We have further knocked down *Msl1* in cultured keratinocytes and observed the reduced H4K16ac levels. In the revision, we have now performed additional qPCR measurement of MOF targets, *Abca11*, *Opa1* and *Scp2*, and documented their downregulation in *Msl1* KD keratinocytes (Fig. 4b). These data suggest that *Msl1* is required to maintain the level of H4K16ac and the expression of selected MOF targets. This result is unexpected, and we are in the process of generating *Msl1* cKO to examine 1) whether *Msl1* cKO alone recapitulate MOF cKO phenotypes; or 2) whether deletion of both *Kansl1* and *Msl1* is required for recapitulating MOF cKO phenotypes; 3) examine whether MSL and NSL occupy different genomic regions and regulate different genes in the skin.

To avoid any confusion, we revised the abstract to simply state that our study shows the requirement of MOF but not *Kansl1* for skin development. We have also stated that future study of *Msl1* cKO will be needed in Discussion.

We hope that the reviewer agrees that generating *Msl1* KO and studying its function are beyond the scope of our current study, and our current results provide important and novel insights for MOF functions in mammalian skin and are suitable for publication at *Nature Communications*.

To address reviewer's request about the expression of NSL components, we used our newly generated scRNAseq datasets at E15.5, which recovered more than 35,000 cells from control and MOF cKO skin. As shown in the figure below, major components of the NSL complex, including *Kansl1*, *Kansl2*, *Kansl3*, *Wdr5*, *Phf20*, *Mcrs1*, *Hcfc1* and *Ogt*, are universally expressed in all epithelial cells, including epithelial cells from MOF cKO. Because this study doesn't focus on whether NSL and/or MSL mediates MOF functions, we did not put these results in the revised manuscript. We hope to use these data in our next publication.

NSL has been shown to be responsible for acetylation of H4K5 and H4K8, but due to lack specific antibodies to those acetylation marks in mouse, we could not perform the relevant experiment to show loss of those marks.

2. The major part of the paper is the analysis of MOF cKO in the epidermis using the Cre- recombinase driven by Keratin 14 promoter. The analysis is then done using cells isolated at E16.5. The major problem with these experiments is that this is examining the end stage of the effects of loss of MOF function. By this point the cells are no longer proliferating (no BrdU labelling) and are either dead, dying or senescent. The epidermis has high levels of activated Caspase 3, around 25-30 cells per field (Extended data fig 2. How many cells in a field? Is the field shown in (c)? Is this E16.5 as implied by the results section?). Activated Caspase3 only marks a small proportion of apoptotic cells as this is a short-lived part of the cascade. So, it is likely that a very high proportion of the cells have undergone apoptosis as soon as the H4K16ac mark dropped below a critical threshold. The cell nuclei although difficult to see at this magnification appear to have clumped chromatin. This suggests the phenotype in these cells resemble the phenotype reported for the germline deletion (Thomas MCB 2008). Overall, the remaining part of the paper is using genomics techniques that by and large are being used to study the end point of apoptosis. If this is just a cell-autonomous lethal phenotype, then it is not surprising that it resembles other cell-autonomous lethal phenotype, for example loss of electron transport chain complexes.

We thank the reviewer's careful comments. Indeed, a major issue for studying MOF cKO is the timing of investigation. Following the reviewer's suggestion, we have generated new scRNAseq datasets from 2 pairs of control and MOF cKO samples at E15.5, when the depletion of H4K16ac is ~70% (Fig. 1a) and when phenotypical defects just began to emerge but not too severe. For example, epidermal integrity was largely intact (Fig. 2a), beta4-integrin was still localized to the basal side of the basal cells (Fig. 2b), epithelial cells were still in the active cycle, shown by universal Ki67 signals in the basal cells (Extended Data fig. 3a), apoptosis was mild (Extended Data fig. 3c). In

comparison to 2.6-4.9% of epithelial cells recovered from E16.5 MOF cKO, we recovered 9.7-14.8% of epithelial cells in E15.5 MOF cKO. The newer 10x v3 kit used for generating the new libraries also has a higher capture rate for mRNA, which allows us to sequence more transcripts per cell. In addition, dermal cells from control and MOF cKO samples at E15.5 clustered together in a completely overlapping manner, different from their E16.5 counterparts. These data indicate that dermal cells have not responded to epidermal defects caused by the deletion of MOF at E15.5. Finally, we fortuitously profiled a complete MOF cKO (cKO5) and a mosaic MOF cKO (cKO6) at E15.5. As shown in new Fig. 3b, epithelial cells of cKO5 were completely clustered away from control epithelial cells. However, 78% of epithelial cells of cKO6 clustered with control epithelial cells whereas 21% clustered with cKO5. The mosaic pattern was further validated by H4K16ac IF signals from the same skin (Extended Data fig. 4e). If MOF cKO simply causes cell death but not specific changes in the transcriptome, one would expect to observe a continuum of single cells, originating from control cell clusters and spreading to cKO cell clusters, reflecting a gradual change in the transcriptome and cell state. In contrast to this prediction, we observed a bimodal distribution pattern of cKO6 cells – either clustered with control cells or clustered with cKO5 cells without transient cells between these clusters (Fig. 3b). These results collectively argue that the loss of MOF leads to a specific change in cellular transcriptome. Together, these technical improvements have significantly enhanced our ability to detect direct response of the transcriptome to the loss of MOF and H4K16ac while minimizing secondary effects. Notably, with this new dataset, we detected even more robust downregulation of nuclear-encoded mitochondrial genes and, to a lesser extent, genes involved in cell cycle, splicing and DNA repair in both basal and suprabasal epidermal cells (Fig. 3c-d), similar to what we learned from E16.5 data. We also detected a few ciliary genes in both epidermis and hair germ of control, and their expression, as expected, is also downregulated in MOF cKO at E15.5. Together, these data have provided additional evidence that MOF-regulated gene expression directly controls mitochondrial genes, including the ETC, and ciliary genes.

3. The authors use scRNA-seq to show that in the MOF KO dermis inflammatory gene expression is up, which is not surprising given the extent of apoptosis in this tissue and mitochondrial gene expression is down. If the cells are not proliferating, and probably dying, it is hardly surprising that their energy requirements are reduced. Likewise, it is not surprising that other cellular functions are also compromised as a result, including ciliary gene expression. I note that p21, a p53 target gene regulating cell cycle arrest is one of the significantly upregulated genes. The authors have not shown that these are specific MOF-regulated genes rather than just a non-specific consequence of major disruption to chromatin organization.

We thank the reviewer's comments. As stated in our response to point #2, we have generated 4 new scRNAseq datasets from E15.5 control and MOF cKO samples. At this earlier time point, the phenotype was mild and H4K16ac levels were reduced to ~30% of control cells (Fig. 1a). Specific to address the reviewer's comment, the dermal cell clusters of E15.5 samples did not differ between control and MOF cKO, unlike our

E16.5 scRNAseq data. Consistent with this observation, when we performed differential gene expression analysis, we detected only 33 upregulated genes and 37 downregulated genes in the dermal cells of E15.5 MOF cKO. This contrasts with 234 upregulated genes and 305 downregulated genes in the dermal cell of E16.5 MOF cKO. Finally, not only were E15.5 epithelial cells still in active cell cycle, as indicated by universal Ki67 signals (Extended Data fig. 3a). Cell cycle analysis of E15.5 scRNAseq data also confirmed the presence of G1, S, G2/M cells in both control and MOF cKO cells (Extended Data fig. 4c). Thus, our E15.5 datasets captured the perturbed transcriptome at the early time point and more likely reflect the direct consequence of the loss of MOF and H4K16ac.

Furthermore, because of our enhanced detection of transcripts, we now detected a few ciliary genes, including B9d1, Cfap36, Ift74 and Kif3a, all of which were significantly downregulated in MOF cKO at E15.5, consistent with our E16.5 bulk RNA-seq results. These gene expression changes were also supported by our new analysis of primary cilia in epidermis and hair germ (Fig. 6).

For the upregulated p21, it could be due to activated p53, as pointed out by the reviewer. However, p21 is also a canonical target of p63, a master transcription factor of epithelial cells, which shares the same binding motif to p53 (Yang et al. Molecular Cell 1998). We note, however, MOF and H4K16ac are known to promote gene expression. Thus, we focus on downregulated genes for identifying their targets in MOF cKO. For upregulated genes, we generally consider them as the secondary effect.

4. The authors then examine the location of MOF and various histone marks in the genome, including H4K16ac. As the authors state this really just confirms published data. This is, however, something of a missed opportunity. While H4K16ac is enriched at TSS it must also be widely distributed throughout the genome. Studies have shown that somewhere between 10-30% of nucleosomes are decorated with H4K16ac (Hansen NatComm 2019), so clearly the peaks shown in this work are just enrichment. Since the authors have MOF KO cells (ie. H4K16ac deficient) they could have use these to determine the real level of H4K16ac and MOF across the genome rather than use peak calling, which more or less assumes most intergenic regions are negative. The authors claim there is a specific function of MOF in regulating Rfx2 target genes. They find 4366 MOF binding peaks at promoters, which overlap with 47% of Rfx2 bound genes. Is this statistically significant? 4366 promoters are a substantial percentage of the expressed genes in a cell and given that the peak calling has a rather arbitrary cut-off, since there is no negative control in this experiment, so it is entirely possible that any group of 300 randomly selected expressed genes would show the same enrichment for MOF at their promoters. Also, the number of repeats of these experiments is not stated in the results or methods section.

We thank the reviewer's suggestion. However, it is technically difficult to determine the level of MOF and H4K16ac in MOF cKO epithelial cells, as suggested by the reviewer. As pointed out by the reviewer, E16.5 epithelial cells (with >80% depletion of H4K16ac)

are dying. Now we have carefully examined E15.5 epithelial cells, which show ~70% depletion of H4K16ac, and showed strongly perturbed transcriptome. First, it's challenging to isolate enough E15.5 epithelial cells for the proposed MOF or H4K16ac Cut & Run, which usually require at least 100,000 cells (our current results were obtained from millions of control cells). Second, even if we managed to get these cells from multiple cKO embryos, the question remains whether they're truly depleted of H4K16ac. It is very difficult to control the depletion of MOF and H4K16ac in mouse models. To best examine genomic occupancy of MOF and H4K16ac, we believe the best approach is to establish a cell culture system, similar to most epigenetic studies, to get enough cells and also control the depletion of MOF and H4K16ac. This type of experiments is very different than genetic mouse models, which are most useful to interrogate the functions of epigenetic mechanisms on cell lineage formation and maintenance, as we have done in this study.

The relationship between MOF and Rfx2 was discovered from multiple lines of evidence. First, Rfx2 and Rfx3 motifs were identified as the most significantly enriched motifs at the promoters of downregulated ciliary genes in MOF cKO. To the best of our knowledge, there has never been reported before that ciliary genes are broadly downregulated upon the deletion of an epigenetic factor. Thus, the unusual downregulation prompted us to examine primary cilia in MOF cKO. Second, we performed TF motif search in all MOF occupied promoters, and we did not recover Rfx motifs. It is because not all genes with MOF occupancy at their promoters are downregulated. Moreover, not all downregulated genes that have MOF occupancy in their promoters share the same transcriptional regulation. For example, although hundreds of nuclear-encoded mitochondrial genes are downregulated, we did not detect strong enrichment for TFs that may regulate these genes at their promoters. This suggests that these nuclear-encoded mitochondrial genes are driven by a diverse pool of TFs, as one may expect. In comparison, the large number of downregulated ciliary genes in MOF cKO are controlled by Rfx2, determined by motif search results and confirmed by Rfx2 Cut & Run results, in the skin. These data thus provide additional evidence that the function of Rfx TFs is uniquely compromised in MOF cKO. Third, a majority (79%) of Rfx2 Cut&Run peaks fall on gene promoter, reflecting its strong preference to promoters. To date, only a few well-studied TFs, such as ETS, Sp1 and YY1, prominently regulate gene promoters. Thus, the discovery of Rfx2 as a TF with a strong preference to the promoter and working together with MOF to control ciliary gene expression in the skin is an important step forward to further understand the interplay between MOF, an important epigenetic regulator, and transcription factors at gene promoters.

We stated the number of repeats for all the genomic data in the method section.

Reviewer #3 (Remarks to the Author):

In this paper, "MOF-mediated Histone H4 Lysine 16 Acetylation Governs Mitochondrial and Ciliary Function By Controlling Gene Promoters," Wang et al. have identified the

histone acetyltransferase, MOF, and its corresponding marker H4K16ac control the mitochondria function and cilia formation to regulate the skin epidermis and hair follicle development. It is hypothesized that MOF and H4K16ac play a fundamental role in tissue development by regulating cellular functions and lineage differentiation during tissue development. In brief, the authors used the conditional depletion of MOF in epidermal basal cells at the embryonic stage. Loss of the H4K16ac specific in the basal cells upon the MOF cKO, they found that several epidermis defect phenotypes and barrier function leads to perinatal death. The authors used the scRNAseq, or bulk RNAseq, combined with the MOF and histone mark Cut and Run results, they identified the mitochondria and primary cilia genes are regulated by MOF-H4K16ac. Moreover, they used genetic deletion of QPC, one of the subunits of the mitochondria electron transport chain (ETC) complex, to phenocopy the epidermis development defect similar to MOF cKO. Overall, the authors reached three major conclusions: 1) NSL complex does not participate in the MOF-mediated regulation of tissue development 2) MOF-H4K16ac changed chromatin accessibility and regulated genes are important for mitochondria function and primary cilia formation. 3) Role of MOF-regulation within mitochondria is vital for epidermal differentiation and hair follicle formation. The results clearly support the conclusions; however, minor additions may need to be considered for a more convincing argument.

We thank the reviewer for positive feedbacks. In the revision, we have addressed all concerns raised by the reviewer and significantly improved our study.

Major points:

1. Is it being suggested that MSL is the complex which mediates MOF function during mammalian tissue development? The argument against NSL is clear, as the authors found no major differences in mammalian development with a cKO of *Kansl1*, an important component of NSL. However, there is no evidence in support of the MSL complex, so the initial question of “whether the function of MOF is mediated by MSL or NSL” is still open. Does MSL mutant mouse also have similar phenotype? My suggestion would be to show that MSL is important for MOF function during development. In figure 1c, the IF image of *Msl1* knockdown by shRNA in keratinocyte is also not clear.

We apologize for the confusion due to the wording of our manuscript. Our primary goal was to study the function and mechanism of MOF in the skin. We were also aware of the functional importance and versatility of the NSL complex, based on several prominent publications (Chatterjee et al., Cell 2016; Li et al., Molecular Cell 2009; and reviewed by Sheikh et al., EMBO Rep 2019). We generated both MOF and *Kansl1* cKO models with the same *Krt14-Cre*. Surprisingly, we observed very strong defects in MOF cKO whereas *Kansl1* cKO is largely intact. The effective deletion of *Kansl1* is well documented by our genotyping, qPCR and RNAseq. The lack of phenotype in *Kansl1* is also consistent with intact H4K16ac levels, determined by both Western blot and IF staining, and lack of transcriptome changes determined by RNAseq. Thus, our data suggest that *Kansl1* is not required for mouse embryonic skin development. We have further knocked down *Msl1* in cultured keratinocytes and observed reduced H4K16ac

levels, supporting the notion that MSL is required to maintain the levels of H4K16ac. We remade Figure 1c to include more cells to clearly show reduced H4K16ac in Msl1 shRNA knockdown keratinocytes. In the revision, we have performed additional qPCR measurement of MOF targets, Abca11, Opa1 and Scp2, and documented their downregulation in Msl1 KD keratinocytes (Fig. 4b). These existing and new data suggest that Msl1 is required to maintain the level of H4K16ac and the expression of selected MOF targets. This result is unexpected, and we are in the process of generating Msl1 cKO to examine 1) whether Msl1 cKO alone recapitulate MOF cKO phenotypes; or 2) whether deletion of both Kansl1 and Msl1 is required for recapitulating MOF cKO phenotypes; 3) examine whether MSL and NSL occupy different genomic regions and regulate different genes in the skin. However, we do not have Msl1 fl/fl mice, and we believe these studies are best suited for another study in the future.

To avoid any confusion, we revised the abstract to simply state that our study shows the requirement of MOF but not Kansl1 for skin development. We have also stated that future study of Msl1 cKO will be needed in Discussion.

We hope that the reviewer agrees with us that generating Msl1 KO and studying its function are beyond the scope of our current study, and our current results provide important and novel insights for MOF functions in mammalian skin and are suitable for publication at *Nature Communications*.

2. Were there any functional analyses done for MOF function in primary cilia? Especially since the cKO QPC does not address some of the observed phenotypes from the MOF cKO. It seems that the function of MOF in this context was overlooked in addition to role of MOF in primary cilia. Also, it lacks evidence to show that the MOF cKO-primary cilia axis affects skin development.

We thank the reviewer for the important question. We provided evidence that MOF binds to promoters of ciliary genes, and a large number of ciliary genes are down-regulated in MOF cKO. These data suggest that MOF and H4K16ac are crucial for the expression of ciliary genes. Furthermore, primary cilia are also significantly reduced (new Fig. 6f-g), further corroborating the importance of MOF-mediated gene expression control for primary cilia formation. Because several recent papers, listed below and cited in our manuscript, have demonstrated the importance of primary cilia for epidermal differentiation, hedgehog signaling and hair follicle development, respectively. Notably, these defects were also observed in MOF cKO. Based on these results, we felt that we could not provide additional new insights into the function of primary cilia beyond the connection between MOF and ciliary gene expression. In contrast, MOF's function, through the regulation of ETC genes, is unknown not only for skin development but also for mammalian development in general. Therefore, we committed our resources to study whether genetic ablation of the ETC, through deletion of a critical subunit of Complex III – QPC, causes skin development defects. We further compared MOF cKO and QPC cKO defects with scRNAseq and revealed many commonly changed genes. We list several previous publication on the function of primary cilia in skin development below:

1. A role for the primary cilium in Notch signaling and epidermal differentiation during skin development. *Cell*, 2011, 145: 1129.
2. Intraflagellar transport 27 is essential for hedgehog signaling but dispensable for ciliogenesis during hair follicle morphogenesis. *Development*, 2015, 142: 2194.
3. Role of epidermal primary cilia in the homeostasis of skin and hair follicles. *Development*, 2011, 138: 1675.

3. The authors performed the bulk RNAseq by FACS purification cells. Can authors show the sorting profiles? Does the MOF cKO affect epithelial markers expression?

To sort epithelial cells, we breed Krt14-H2bGFP transgenic mice, labeling all epithelial cells at E16.5, with MOF cKO or control. By sorting H2bGFP+ cells from E16.5 skin, we obtained total epithelial cells for bulk RNAseq. A representative sorting profile is now presented in Extended Data fig. 8a. Based on our RNAseq data and IF staining, it is confirmed that several basement membrane markers are reduced whereas epithelial fate markers, such as Krt5, Krt14 and p63 are not changed. Therefore, our sorting strategy was not affected by MOF cKO status.

4. Is the H3K4me3 the global reduction or mitochondria/primary cilia genes specifically in MOF cKO?

This is a great question. To address this question, H3K4me3 Cut & Run will be needed. However, we haven't fully optimized our protocol to enable the assay with 5~10K sorted cells (the number of MOF cKO epithelial cells we can sort from an embryo). We note, however, that open chromatin signatures are correlated with H3K4me3. In our analysis of open chromatin regions, we found that MOF targeted promoters show significant reduction of open chromatin signatures, in contrast to the promoters of non-targeted epithelial marker genes (Fig. 8b). Therefore, it is likely that H3K4me3 is more depleted in MOF targeted promoters. A recent study has shown that H3K4me3 is required for transcriptional output by promoting pol II pause-release (Wang et al., *Nature* 2023). Because less than 1,500 genes are downregulated in our scRNAseq from E15.5 and E16.5 or bulk RNAseq from E16.5, it suggests that these genes more likely have reduced H3K4me3 at their promoter in MOF cKO. We discussed this point, and will perform additional experiments to interrogate how MOF cKO and the loss of H4K16ac selectively reduce H3K4me3 levels at some promoters.

Minor points

1. The authors should include detailed information about the quantification method in the material and method. It is also unclear in the figure; What is the unit in quantifying immunostaining?

We included in the method for detailed information about data quantification.

2. It would be beneficial for the conclusion that there is a reduction in HF formation in MOF cKO mice (Figure 2a), to have quantitative data like what was done in Figure 7b.

We appreciate the reviewer's suggestion. However, MOF cKO, unlike QPC cKO, has severe defects in the formation of basement membrane, which is critical not only for basal cell adhesion but also for epithelial-dermal interactions. In regions where the basement membrane is formed, hair follicle induction can be robustly detected as we showed in new Extended Data fig. 3f (originally Extended Data fig. 2e-f). However, in regions where the basement membrane is severely compromised such that epidermis is detached from the underlying dermal cells, hair follicles can no longer develop into more advanced stages. Therefore, we couldn't quantify the stages as we have done for QPC cKO, which did not have defects in the basement membrane. In the revision, we added a discussion to distinguish the differences in the formation of basement membrane between MOF cKO and QPC cKO and the impact on hair follicle development.

3. What is the age of *Knasl1* cKO in figure 1d-f?

All *Knasl1* cKO were analyzed at P0.5. This is now labelled on the figures and stated in the figure legend.

4. In line 145, the authors mention "E18.5 epidermal detachment became widespread, and the reduction of basal cells..." How do the authors get the conclusion about the reduction of basal cells from figure 2a? Are there any markers staining and quantification results to indicate that reduce the basal cells in MOF cKO epidermis at E18.5?

We performed *Krt5* staining at E18.5 and found that some region has largely lost *Krt5*+ basal cells. Now we show an example in Extended Data fig 2b.

5. In figure 2e, the E-cad and α -Cat cannot clearly see the distribution of these two markers change in control and MOF KO.

We thank the reviewer's careful observation. We did have a version with both channels shown separately. However, due to the limited space, we only used channel-combined images. The point here is to show that E-Cadherin is generally excluded from the basal side of the basal cells in control sample, while α -Catenin locates around the basal cells. As a result, the basal side of the basal cells in control is only stained with α -Catenin signals (red). In contrast, since MOF cKO basal cells lost this polarity, the whole cell

membrane became yellow with overlapping E-cad and a-cat signals, caused by mislocation of E-Cadherin to the basal side. We added a few arrowheads to the images to point out the differences.

6. In line 248, "MOF and H4K16ac have been implicated in the regulation of "housekeeping" genes...." Please cite the references in this sentence.

We now cited two papers, both were from studies in drosophila.

7. In figure 4e, what is the "HG" in the ctrl skin? The primary cilia are enriched in the HG in ctrl skin. Is there any HG in the MOF cKO skin?

We apologize for the confusion. HG refers to hair germ, an early stage of hair follicle development. They can be distinguished by the cell aggregation patterns from the planar view. In the revision, we did new analysis for the defects of primary cilia in MOF cKO skin, and quantified primary cilia in hair germ and epidermal progenitor cells separately as shown in new Fig. 6f-g. In both compartments, primary cilia are significantly reduced in MOF cKO skin, consistent with the widespread downregulation of numerous ciliary genes in MOF cKO.

8. In figure 6, Can authors show the profiles of the MOF target genes such as mitochondria and primary cilia genes in control and MOF cKO? It would be much clearer to see the chromatin accessibility changes between control and cKO skin.

We thank the reviewer's suggestion to strengthen our presentation. In the revision, we show two examples of reduced open chromatin signals on MOF targeted genes in control and MOF cKO epidermis in new Fig. 8d. We also show one example of intact open chromatin signals on a gene, not targeted by MOF, in new fig. 8e.

REVIEWERS' COMMENTS

Reviewer #1 (Remarks to the Author):

The addition of solid scRNAseq data at an earlier time point, and the in vivo validations of the phenotypes strongly strengthened the study.

The revisions have addressed all my major concerns and I believe that the article is now ready for publication, with one minor suggestion for consideration.

My only minor comment was for figure 8, with respect to changes in chromatin accessibility upon MOF KO. While I see clear differences in Figure 8 a and b (but not in c), as discussed by the authors, the examples of individual genes shown in 8d and 8e were difficult to assess. Is there another way to present this data, which would make it easier to catch the differences? For instance, integrate the signal over promoter and present it as a bar, so the fractional change is more apparent? Or find better examples that were changed more dramatically? Or overlay the Ctrl vs KO ATAC-seq profiles so the differences are more easily observed. This seems like an unfortunate less than convincing ending of a paper that is otherwise a real tour de force.

I commend the authors on a thorough and exciting study and now recommend it for publication.

Reviewer #2 (Remarks to the Author):

The authors have undertaken a considerable number of additional experiments. In particular, the authors have extended their study to include embryos at E15.5. This is important because at E15.5 the effects of the knock-out are only just beginning to be felt, so the secondary downstream effects of the loss of a major epigenetic regulator are less apparent.

My criticisms of the first version of the manuscript have been fully addressed in the revised version.

Reviewer #3 (Remarks to the Author):

The authors fully addressed my previous concerns. Based on the improvements made, I believe the paper is now suitable for publication. Congratulations to the authors on their outstanding work.

REVIEWERS' COMMENTS

Reviewer #1 (Remarks to the Author):

The addition of solid scRNAseq data at an earlier time point, and the in vivo validations of the phenotypes strongly strengthened the study.

The revisions have addressed all my major concerns and I believe that the article is now ready for publication, with one minor suggestion for consideration.

My only minor comment was for figure 8, with respect to changes in chromatin accessibility upon MOF KO. While I see clear differences in Figure 8 a and b (but not in c), as discussed by the authors, the examples of individual genes shown in 8d and 8e were difficult to assess. Is there another way to present this data, which would make it easier to catch the differences? For instance, integrate the signal over promoter and present it as a bar, so the fractional change is more apparent? Or find better examples that were changed more dramatically? Or overlay the Ctrl vs KO ATAC-seq profiles so the differences are more easily observed. This seems like an unfortunate less than convincing ending of a paper that is otherwise a real tour de force.

I commend the authors on a thorough and exciting study and now recommend it for publication.

We thank the reviewer's support, and we have revised Figure 8 to overlay the ATAC signals from control and cKO samples and show the differences more clearly.

Reviewer #2 (Remarks to the Author):

The authors have undertaken a considerable number of additional experiments. In particular, the authors have extended their study to include embryos at E15.5. This is important because at E15.5 the effects of the knock-out are only just beginning to be felt, so the secondary downstream effects of the loss of a major epigenetic regulator are less apparent.

My criticisms of the first version of the manuscript have been fully addressed in the revised version.

We thank the reviewer's support to our study.

Reviewer #3 (Remarks to the Author):

The authors fully addressed my previous concerns. Based on the improvements made, I believe the paper is now suitable for publication. Congratulations to the authors on their outstanding work.

We thank the reviewer's support to our study.